# Neural varifolds: an aggregate representation for quantifying geometry of point clouds

## Abstract

Point clouds are popular 3D representations for real-life objects (such as in LiDAR and Kinect) due to their detailed and compact representation of surface-based geometry. Recent approaches characterise the geometry of point clouds by bringing deep learning based techniques together with geometric fidelity metrics such as optimal transportation costs (e.g. Chamfer and Wasserstein metrics). In this paper, we propose a new surface geometry characterisation within this realm, namely a neural varifold representation of point clouds. Here the surface is represented as a measure/distribution over both point positions and tangent spaces of point clouds. The varifold representation not only helps to quantify the surface geometry of point clouds through the manifold-based discrimination, but also subtle geometric consistency on the surface due to the combined product space. This study proposes neural varifold algorithms to compute varifold norm between two point clouds using neural networks on point clouds and their neural tangent kernel representations. The proposed neural varifold is evaluated on three different tasks – shape classification, shape reconstruction and shape matching. Detailed evaluation and comparison to the state-of-the-art methods demonstrate that the proposed versatile neural varifold is superior in shape classification particularly for limited data and is quite competitive for shape reconstruction and matching.

## 1 Introduction

Point clouds are preferred in more and more applications including computer graphics, autonomous driving, robotics and augmented reality. However, manipulating/editing point clouds data in their raw form is rather cumbersome. Neural networks have made breakthroughs in a wide variety of fields ranging from natural language processing to computer vision, albeit its success is mainly on audio, texts and images in which there is an underlying Euclidean grid structure. Three-dimensional (3D) data in general do not have underlying grid structures, such that convolution operations on Euclidean grid are not applicable. Geometric deep learning and its variants have addressed technical problems of translating neural networks on non-Euclidean data (Bronstein et al., 2017). With advanced graph theory and harmonic analysis, convolutions on non-Euclidean data can be defined in the context of spectral (Bruna et al., 2014; Defferrard et al., 2016) or spatial (Monti et al., 2017; Wang et al., 2019) domains. Although geometric deep learning on point clouds has successfully achieved top performance in shape classification and segmentation tasks, capturing subtle changes in 3D surface remains challenging due to the unstructured and non-smooth nature of point clouds. A possible direction to learn subtle changes on 3D surface adopts some concepts developed in the field of theoretical geometric analysis. In other words, deep learning architectures might be improved by incorporating theoretical knowledge from geometric analysis. In this work, we introduce concepts borrowed from geometric measure theory, where representing shapes as measures or distributions has been instrumental.

Geometric measure theory has been actively investigated by mathematicians; however, its technicality may have hindered its popularity and its use in many applications. Geometric measure-theoretic concepts have recently been introduced to measure shape correspondence in non-rigid shape matching (Vaillant & Glaunès, 2005; Charon & Trouvé, 2013; Hsieh & Charon, 2021) and curvature estimation (Buet et al., 2017; 2022). We introduce the theory of varifolds to improve learning representation of 3D point clouds. An oriented $d$-varifold is a measure over point positions and oriented

tangent $k$-planes, i.e. a measure on the Cartesian product space of $\mathbb{R}^n$ and the oriented Grassmannian manifold $\tilde{G}(d, n)$. Varifolds can be viewed as generalisations of $d$-dimensional smooth shapes in Euclidean space $\mathbb{R}^n$. The varifold structure not only helps to better differentiate the macro-geometry of the surface through the manifold-based discrimination, but also the subtle singularities in the surface due to the combined product space. Varifolds provide representations of general surfaces without parameterization, and not only they can represent consistently point clouds that approximate surfaces in 3D, but they are also scalable to arbitrary surface discretisation (e.g., meshes). In this study, we use varifolds to analyze and quantify the geometry of point clouds.

**Our contributions:**

- Introduce neural varifold as learning representation of point clouds. Varifold representation of 3D point clouds coupling space position and tangent planes can provide both theoretical and practical analyses of the surface geometry.

- Propose two algorithms to compute varifold norm between two point clouds using neural networks on point clouds and their neural tangent kernel representations. The reproducing kernel Hilbert space of varifold is computed by the product of two neural tangent kernels of positional and Grassmannian features of point clouds. Neural varifold can take advantage of the expressive power of neural networks as well as varifold representation of point clouds.

- Neural varifold can be used to evaluate shape similarity between point clouds on various tasks including shape classification with limited data, shape reconstruction and shape matching.

## 2 RELATED WORKS

**Geometric deep learning on point clouds.** PointNet is the first pioneering work on point clouds. It consists of a set of fully connected layers followed by symmetric functions to aggregate feature representations. In other words, PointNet is neural networks on a graph without edge connections. In order to incorporate local neighbourhood information with PointNet, PointNet++ (Qi et al., 2017b) applied PointNet to individual patches of the local neighbourhood, and then stacking them together. PointCNN (Li et al., 2018) further refined the PointNet framework with hierarchical $\mathcal{X}$-Conv which calculates inner products of $\mathcal{X}$-transformation and convolution filters of point clouds. Dynamic graph CNN (DGCNN) (Wang et al., 2019) adopted the graph neural network framework to incorporate local neighbourhood information by applying convolutions over the graph edges and dynamically updating graph for each layer. Furthermore, the tangent convolution architecture (Tatarchenko et al., 2018) incorporated 3D surface geometry by projecting point clouds on local tangent plane, and then applying convolution filters.

**Varifold.** Theoretical understanding of surface geometry has been studied in the context of geometric measure theory and calculus of variations. Various tools from geometric measure theory can be used to study surface geometry, e.g., currents (Vaillant & Glaunès, 2005), varifolds (Charon & Trouvé, 2013; Buet et al., 2017; 2022) and normal cycles (Roussillon & Glaunès, 2019). Despite of its potential usage for many applications, a few studies have explored real-world applications in the context of non-rigid surface registration (Charon & Trouvé, 2013).

## 3 VARIFOLD REPRESENTATIONS FOR POINT CLOUDS

The notion of varifold comes from geometric measure theory in the context of finding a minimal surface spanning a given closed curve in $\mathbb{R}^3$, which is known as Plateau's problem (Allard, 1975). Let $\Omega \subset \mathbb{R}^n$ be an open set. A general oriented $d$-varifold $V$ on $\Omega$ is a non-negative Radon measure on the product space of $\Omega$ with the oriented Grassmannian $\tilde{G}(d, n)$. In this study, we focus on a specific class of varifolds, the rectifiable varifolds, that are concentrated on $d$-rectifiable sets and can represent non-smooth surfaces such as 3D cubes.

**Definition 3.1 (Rectifiable oriented $d$-varifolds)** *Let $\Omega \subset \mathbb{R}^n$ be an open set, $X$ an oriented $d$-rectifiable set, and let $\theta$ be a non-negative measurable function with $\theta > 0$ $\mathcal{H}^d$-almost everywhere in $X$. The rectifiable oriented $d$-varifold $V = v(\theta, X)$ in $\Omega$ is the Radon measure on $\Omega \times \tilde{G}(d, n)$*

*defined by $V = \theta\mathcal{H}^d_{X \cap \Omega} \otimes \delta_{T_x X}$, i.e.,*

$$\int_{\Omega \times \tilde{G}(d,n)} \phi(x,T)\mathrm{d}\mu(x,T) = \int_X \phi(x,T_x X)\theta(x)\mathrm{d}\mathcal{H}^d(x), \ \ \forall \phi \in C_0(\Omega \times \tilde{G}(d,n)),$$

*where $C_0$ denotes the class of continuous functions vanishing at infinity.*

The mass of a $d$-rectifiable varifold $V = v(\theta, X)$ is the measure $\|V\| = \theta\mathcal{H}^d_X$. The non-negative function $\theta$ is usually called multiplicity. We assume in the rest of the paper that $\theta = 1$ for simplicity.

Various metrics and topologies can be defined on the space of varifolds. The mass distance defined as follows is a possible choice for a metric:

$$d_{\mathrm{mass}}(\mu,\nu) = \sup\left\{\left|\int_{\Omega \times \tilde{G}(d,n)} \phi\mathrm{d}\mu - \int_{\Omega \times \tilde{G}(d,n)} \phi\mathrm{d}\nu\right|, \phi \in C_0(\Omega \times \tilde{G}(d,n)), \|\phi\|_\infty \leq 1\right\}. \quad (1)$$

However, the mass distance is not well suited for point clouds. For example, given the two varifolds associated with Dirac masses $\delta_\varepsilon$ and $\delta_0$, their distance remains bounded away from 0 as it is always possible to find a test function $\phi$ such that $|\phi(0) - \phi(\varepsilon)| = 2$, regardless of how close the two points are. The 1-Wasserstein distance is not a more suitable choice in our context for it cannot compare two varifold measures with different mass. For example, given two Dirac masses $(1 + \varepsilon)\delta_0$ and $\delta_0$, the 1-Wasserstein distance between them goes to infinity as $\varepsilon|\phi(0)| \to \infty$.

**Definition 3.2 (Bounded Lipschitz distance)** *Being $\mu$ and $\nu$ two varifolds on a locally compact metric space $(X, d)$, we define*

$$d_{\mathrm{BL}}(\mu,\nu) = \sup\left\{\left|\int_{\Omega \times \tilde{G}(d,n)} \phi\mathrm{d}\mu - \int_{\Omega \times \tilde{G}(d,n)} \phi\mathrm{d}\nu\right|, \phi \in C_0^1(\Omega \times \tilde{G}(d,n)), \ \|\phi\|_{Lip} \leq 1, \|\phi\|_\infty \leq 1\right\}.$$

$$(2)$$

The bounded Lipschitz distance (flat distance) can handle both problems, we refer for more details to Piccoli & Rossi (2016) and the references therein. Although the bounded Lipschitz distance $d_{\mathrm{BL}}$ can provide theoretical properties for comparing varifolds, in practice, there is no straightforward way to numerically evaluate it. Instead, kernel regime has been used to evaluate varifolds numerically (Charon & Trouvé, 2013; Hsieh & Charon, 2021).

**Proposition 3.3 (Hsieh & Charon (2021))** *Let $k_{\mathrm{pos}}$ and $k_G$ be continuous positive definite kernels on $\mathbb{R}^n$ and $\tilde{G}(d,n)$, respectively. Assume in addition that for any $x \in \mathbb{R}^n$, $k_{\mathrm{pos}}(x, \cdot) \in C_0(\mathbb{R}^n)$. Then $k_{\mathrm{pos}} \otimes k_G$ is a positive definite kernel on $\mathbb{R}^n \times \tilde{G}(d,n)$, and the reproducing kernel Hilbert space (RKHS) $W$ associated with $k_{\mathrm{pos}} \otimes k_G$ is continuously embedded in $C_0(\mathbb{R}^n \times \tilde{G}(d,n))$, i.e., there exists $c_W > 0$ such that for any $\phi \in W$, we have $\|\phi\|_\infty < c_W\|\phi\|_W$.*

Let $\tau_W : W \mapsto C_0(\mathbb{R}^n \times \tilde{G}(d,n))$ be the continuous embedding given by Proposition 3.3 and $\tau_{W^*}$ be its adjoint. Then varifolds can be viewed as elements of the dual RKHS $W^*$. Let $\mu$ and $\nu$ be two varifolds. By the Hilbert norm of $W^*$, the pseudo-metric can be induced as follows

$$d_{W^*}(\mu,\nu)^2 = \|\mu - \nu\|^2_{W^*} = \|\mu\|^2_{W^*} - 2\langle\mu,\nu\rangle_{W^*} + \|\nu\|^2_{W^*}. \quad (3)$$

The above pseudo-metric (since $\tau_{W^*}$ is not injective in general) is associated with the RKHS $W$, and it provides an efficient way to compute varifold by separating the positional and Grassmannian components. Indeed, one can derive a bound with respect to $d_{\mathrm{BL}}$ if we further assume that RKHS $W$ is continuously embedded into $C_0^1(\mathbb{R}^n \times \tilde{G}(d,n))$ (Charon & Trouvé, 2013), i.e.,

$$\|\mu - \nu\|_{W^*} = \sup_{\phi \in W, \|\phi\|_W \leq 1} \int_{\mathbb{R}^n \times \tilde{G}(d,n)} \phi\,\mathrm{d}(\mu - \nu) \leq c_W d_{BL}(\mu,\nu).$$

**Neural tangent kernel.** The recent advances of neural network theory finds a link between kernel theory and over-parameterised neural networks (Jacot et al., 2018; Arora et al., 2019a). If a neural network has a large but finite width, the weights at each layer remain close to its initialisation. Given training data pairs $\{\boldsymbol{x}_i, y_i\}_{i=1}^N$ where $\boldsymbol{x}_i \in \mathbb{R}^{d_0}$ and $y_i \in \mathbb{R}$, let $f(\boldsymbol{\theta}; \boldsymbol{x}_i)$ be a fully-connected neural

network with $L$-hidden layers with inputs $\boldsymbol{x}_i$ and parameters $\boldsymbol{\theta} = \{\boldsymbol{W}^{(0)}, \boldsymbol{b}^{(0)}, \ldots, \boldsymbol{W}^{(L)}, \boldsymbol{b}^{(L)}\}$. Let $d_h$ be the width of the neural network for each layer $h$. The neural network function $f$ can be written recursively as

$$\boldsymbol{f}^{(h)}(\boldsymbol{x}) = \boldsymbol{W}^{(h)}\boldsymbol{g}^{(h)}(\boldsymbol{x}) + \boldsymbol{b}^{(h)}, \;\; \boldsymbol{g}^{(h)}(\boldsymbol{x}) = \varphi(\boldsymbol{f}^{(h-1)}(\boldsymbol{x})), \quad h = 0, \ldots, L, \tag{4}$$

where $\boldsymbol{g}^{(0)}(\boldsymbol{x}) = \boldsymbol{x}$ and $\varphi$ is a non-linear activation function.

Assume the weights $\boldsymbol{W}^{(h)} \in \mathbb{R}^{d_{h+1} \times d_h}$ and bias $\boldsymbol{b}^{(h)} \in \mathbb{R}^{d_h}$ at each layer $h$ are initialised with Gaussian distribution $\boldsymbol{W}^{(h)} \sim \mathcal{N}(0, \frac{\sigma_\omega^2}{d_h})$ and $\boldsymbol{b}^{(h)} \sim \mathcal{N}(0, \sigma_b^2)$, respectively. Consider training a neural network by minimising the least square loss function

$$l(\boldsymbol{\theta})) = \frac{1}{2} \sum_{i=1}^{N} (f(\boldsymbol{\theta}; \boldsymbol{x}_i) - y_i)^2. \tag{5}$$

Suppose the least square loss $l(\boldsymbol{\theta}))$ is minimised with an infinitesimally small learning rate, i.e., $\frac{d\boldsymbol{\theta}}{dt} = -\nabla l(\boldsymbol{\theta}(t))$. Let $\boldsymbol{u}(t) = (f(\boldsymbol{\theta}(t); \boldsymbol{x}_i))_{i\in[N]} \in \mathbb{R}^N$ be the neural network outputs on all $x_i$ at time $t$, and $\boldsymbol{y} = (y_i)_{i\in[N]}$ be the desired output. Then $\boldsymbol{u}(t)$ follows the evolution

$$\frac{d\boldsymbol{u}}{dt} = -\boldsymbol{H}(t)(\boldsymbol{u}(t) - \boldsymbol{y}), \tag{6}$$

where

$$\boldsymbol{H}(t)_{ij} = \left\langle \frac{\partial f(\boldsymbol{\theta}(t); \boldsymbol{x}_i)}{\partial \boldsymbol{\theta}}, \frac{\partial f(\boldsymbol{\theta}(t); \boldsymbol{x}_j)}{\partial \boldsymbol{\theta}} \right\rangle. \tag{7}$$

If the width of the neural network at each layer goes to infinity, i.e., $d_h \to \infty$, with a fixed training set, then $\boldsymbol{H}(t)$ remains unchanged. Under random initialisation of the parameters $\boldsymbol{\theta}$, $\boldsymbol{H}(0)$ converges in probability to a deterministic kernel $\boldsymbol{H}^*$ – the "*neural tangent kernel*" (i.e., NTK) (Jacot et al., 2018). Indeed, with few known activation functions $\varphi$ (e.g. ReLU), the neural tangent kernel $\boldsymbol{H}^*$ can be computed by a closed-form solution recursively using Gaussian process (Lee et al., 2017; Arora et al., 2019a). For each layer $h$, the corresponding covariance function is defined as

$$\boldsymbol{\Sigma}^{(0)}(\boldsymbol{x}_i, \boldsymbol{x}_j) = \sigma_b^2 + \frac{\sigma_\omega^2}{d_0} \boldsymbol{x}_i \boldsymbol{x}_j^T, \tag{8}$$

$$\boldsymbol{\Lambda}^{(h)}(\boldsymbol{x}_i, \boldsymbol{x}_j) = \begin{bmatrix} \boldsymbol{\Sigma}^{(h-1)}(\boldsymbol{x}_i, \boldsymbol{x}_i) & \boldsymbol{\Sigma}^{(h-1)}(\boldsymbol{x}_i, \boldsymbol{x}_j) \\ \boldsymbol{\Sigma}^{(h-1)}(\boldsymbol{x}_i, \boldsymbol{x}_j) & \boldsymbol{\Sigma}^{(h-1)}(\boldsymbol{x}_j, \boldsymbol{x}_j) \end{bmatrix} \in \mathbb{R}^{2 \times 2},$$

$$\boldsymbol{\Sigma}^{(h)}(\boldsymbol{x}_i, \boldsymbol{x}_j) = \sigma_b^2 + \sigma_\omega^2 \mathbb{E}_{(u,v)\sim\mathcal{N}(0,\boldsymbol{\Lambda}^{(h)})} [\varphi(u)\varphi(v)]. \tag{9}$$

In order to compute the neural tangent kernel, derivative covariance is defined as

$$\dot{\boldsymbol{\Sigma}}^{(h)}(\boldsymbol{x}_i, \boldsymbol{x}_j) = \sigma_\omega^2 \mathbb{E}_{(u,v)\sim\mathcal{N}(0,\boldsymbol{\Lambda}^{(h)})} [\dot{\varphi}(u)\dot{\varphi}(v)]. \tag{10}$$

Then, the neural tangent kernel at each layer $\boldsymbol{\Theta}^{(h)}$ can be computed as follows

$$\boldsymbol{\Theta}^{(h)}(\boldsymbol{x}_i, \boldsymbol{x}_j) = \boldsymbol{\Sigma}^{(h)}(\boldsymbol{x}_i, \boldsymbol{x}_j) + \boldsymbol{\Theta}^{(h-1)}\dot{\boldsymbol{\Sigma}}(\boldsymbol{x}_i, \boldsymbol{x}_j), \;\; \boldsymbol{\Theta}^{(0)}(\boldsymbol{x}_i, \boldsymbol{x}_j) = \boldsymbol{\Sigma}^{(0)}(\boldsymbol{x}_i, \boldsymbol{x}_j). \tag{11}$$

The convergence of $\boldsymbol{\Theta}^{(L)}(\boldsymbol{x}_i, \boldsymbol{x}_j)$ to $\boldsymbol{H}_{ij}^*$ is proven in Theorem 3.1 in Arora et al. (2019a).

### 3.1 NEURAL VARIFOLD COMPUTATION

In this section, we present the kernel representation of varifold on point clouds via neural tangent kernel. We first introduce neural tangent kernel representation of popular neural networks on point clouds (Qi et al., 2017a; Arora et al., 2019a) by computing the neural tangent kernel for position and Grassmannian components, individually.

Given the set of $n$ point clouds $\mathcal{S} = \{s_1, s_2, \ldots, s_n\}$, where each point cloud $s_i = \{p_1, p_2, \ldots, p_m\}$ is a set of points, and $n, m$ are respectively the number of point clouds and points in each point cloud. Consider PointNet-like architecture that consists of $L$-hidden layers fully connected neural network shared by all points. For $(i, j) \in [m] \times [m]$, the covariance matrix $\boldsymbol{\Sigma}^{(h)}(p_i, p_j)$ and neural tangent

kernel $\mathbf{\Theta}^{(h)}(p_i, p_j)$ at layer $h$ are defined and computed in the same way of Equations 8 and 11. Assuming each point $p_i$ consists of positional information and surface normal direction such that $p_i \in \mathbb{R}^3 \times \mathbb{S}^2$, the varifold representation can be defined with neural tangent kernel theory in two different ways. One way is to follow the Charon-Trouvé approach (Charon & Trouvé, 2013) by computing the position and Grassmannian kernels separately. While the original Charon-Trouvé approach uses the radial basis kernel for the positional elements and a Cauchy-Binet kernel for the Grassmannian parts, in our cases, we use the neural tangent kernel representation for both the positional and Grassmannian parts. Let $\{x_i, n_i\} \in p_i$ be a set of positions $x_i \in \mathbb{R}^3$ and its surface normal $n_i \in \mathbb{S}^2$ pairs. The neural varifold representation is defined as

$$\mathbf{\Theta}^{\text{varifold}}_{(i,j)}(p_i, p_j) = \mathbf{\Theta}^{\text{pos}}_{(i,j)}(x_i, x_j) \odot \mathbf{\Theta}^{G}_{(i,j)}(n_i, n_j). \tag{12}$$

We refer the above representation as PointNet-NTK1. As shown in Corollary 3.4, PointNet-NTK1 is a valid Charon-Trouvé type kernel. From the neural tangent theory of view, PointNet-NTK1 in Equation 12 has two infinite-width neural networks on positional and Grassmannian components separately, and then aggregates information from the neural networks by element-wise product of two neural tangent kernels.

**Corollary 3.4** *In the limit of resolution going to infinity, neural tangent kernels $\mathbf{\Theta}^{\text{pos}}$ and $\mathbf{\Theta}^{G}$ are continuous positive definite kernels on positions and tangent planes, respectively. The varifold kernel $\mathbf{\Theta}^{\text{varifold}} = \mathbf{\Theta}^{\text{pos}} \odot \mathbf{\Theta}^{G}$ is a positive definite kernel on $\mathbb{R}^n \times \tilde{G}(d, n)$ and the associated RKHS $W$ is continuously embedded into $C_0(\mathbb{R}^n \times \tilde{G}(d, n))$.*

The other way to define a varifold representation is by treating each point as a 6-dimensional feature $p_i = \{x_i, n_i\} \in \mathbb{R}^6$. In this case, a single neural tangent kernel corresponding to an infinite-width neural network can be used, i.e.,

$$\mathbf{\Theta}^{\text{varifold}}_{(i,j)}(p_i, p_j) = \mathbf{\Theta}^{\text{pos}}_{(i,j)}(\{x_i, n_i\}, \{x_j, n_j\}). \tag{13}$$

We refer it as PointNet-NTK2. Since PointNet-NTK2 does not compute the positional and Grassmannian kernels separately, it is computationally cheaper than PointNet-NTK1. It cannot be associated in the limit with a Charon-Trouvé type kernel, in contrast with PointNet-NTK1, but it remains theoretically well grounded because the explicit coupling of positions and normals is a key aspect of the theory of varifolds that provides strong theoretical guarantees (convergence, compactness, weak regularity, second-order information, etc.). Furthermore, PointNet-NTK2 falls into the category of neural networks proposed for point clouds (Qi et al., 2017a;b) that treat point positions and surface normals as 6-feature vectors, and thus PointNet-NTK2 is a natural extension of current neural networks practices for point clouds.

PointNet-NTK1 and PointNet-NTK2 in Equations 12 and 13 are computing NTK values between two points $p_i$ and $p_j$. The above forms can compute only pointwise-relationship in a single point cloud. However, in many point cloud applications, two or more point clouds need to be evaluated. Given the set of point clouds $\mathcal{S}$, one needs to compute a Gram matrix of size $n \times n \times m \times m$, which is computationally prohibitive in general. In order to reduce the size of the Gram matrix, we aggregate information by summation/average in all elements of $\mathbf{\Theta}^{\text{varifold}}_{(k,l)}$, thus forming an $n \times n$ matrix, i.e.,

$$\mathbf{\Theta}^{\text{varifold}}(s_k, s_l) = \sum_{i \in |s_k|} \sum_{j \in |s_k|} \mathbf{\Theta}^{\text{varifold}}_{(i,j)}(p_i, p_j). \tag{14}$$

Analogous to Equation 3, the varifold representation $\mathbf{\Theta}^{\text{varifold}}$ can be used as a shape similarity metric between two sets of point clouds $s_k$ and $s_l$. The varifold metric can be computed as follows

$$\|s_k - s_l\|^2_{\text{varifold}} = \mathbf{\Theta}^{\text{varifold}}(s_k, s_k) - 2\mathbf{\Theta}^{\text{varifold}}(s_k, s_l) + \mathbf{\Theta}^{\text{varifold}}(s_l, s_l). \tag{15}$$

Furthermore, the varifold representation can be used for shape classification or any regression with the labels on point clouds data. Given training and test point cloud sets and their label pairs $(\boldsymbol{\chi}_{\text{train}}, \boldsymbol{\mathcal{Y}}_{\text{train}}) = \{(s_1, y_1), \dots, (s_l, y_l)\}$ and $(\boldsymbol{\chi}_{\text{test}}, \boldsymbol{\mathcal{Y}}_{\text{test}}) = \{(s_{l+1}, y_{l+1}), \dots, (s_n, y_n)\}$, then neural varifold and its norm can be reformulated to predict labels using kernel ridge regression, i.e.,

$$\boldsymbol{\mathcal{Y}}_{\text{test}} = \mathbf{\Theta}^{\text{varifold}}_{\text{test}}(\boldsymbol{\chi}_{\text{test}}, \boldsymbol{\chi}_{\text{train}})(\mathbf{\Theta}^{\text{varifold}}_{\text{train}}(\boldsymbol{\chi}_{\text{train}}, \boldsymbol{\chi}_{\text{train}}) + \lambda \boldsymbol{I})^{-1} \boldsymbol{\mathcal{Y}}_{\text{train}}, \tag{16}$$

where $\lambda$ is the regularisation parameter.

## 4 EXPERIMENTS

**Dataset and experimental setting.** We evaluate the varifold kernel representations and conduct comparisons on three different tasks: point cloud based shape classification with limited data, point cloud based 3D shape reconstruction and point cloud based shape matching between two different 3D meshes. For ease of reference, we below shorten our proposed neural varifold methods PointNet-NTK1 and PointNet-NTK2 as NTK1 and NTK2, respectively.

To perform shape classification on point cloud data with limited samples, we utilised the Princeton ModelNet benchmark (Wu et al., 2015; Ye et al., 2023). The ModelNet benchmark has two classification tasks: (i) ModelNet10 is a shape classification benchmark for 10 selected classes; and (ii) ModelNet40 is a shape classification benchmark for the whole 40 classes. The number of 1024 points and their corresponding normals for each object were sampled from the original meshes and used for both ModelNet10 and ModelNet40 classification tasks. The proposed neural varifold methods are compared with popular neural networks on point clouds including PointNet (Qi et al., 2017a), DGCNN (Wang et al., 2019) as well as the kernel method Charon & Trouvé (2013). The computation of the neural varifold kernels (NTK1 and NTK2) is based on the *neural tangent* library (Novak et al., 2020). Each method was also trained with different number of training samples varying from 1 to 50. To make the results more consistent, samples were randomly chosen and iterated 20 times with different seeds. Both NTK1 and NTK2 are required to fix the number of layers corresponding to the equivalent finite-width neural networks. NTK1 uses 5 fully connected neural network layers while NTK2 adopts 9 fully connected neural network layers. Each layer consists of MLP, layer normalization and ReLU activation for both NTK1 and NTK2. The shape classification performance on the full ModelNet data is available at Appendix A.2. The criteria used to choose the number of layers and different layer width for both NTK1 and NTK2 are available at Appendix A.4.

For shape reconstruction from point clouds, ShapeNet dataset (Chang et al., 2015) was used. In particular, we followed the data processing and shape reconstruction experiments from Williams et al. (2021), i.e., 20 objects from the individual 13 classes were randomly chosen and used for evaluating the shape reconstruction performance. For each shape, 2048 points were sampled from the surface and used for the reconstruction. Our approach was compared with the state-of-the-art shape reconstruction methods including Neural Splines (Williams et al., 2021), SIREN (Sitzmann et al., 2020) and neural kernel surface reconstruction (NKSR) (Huang et al., 2023). To be consistent with existing point cloud based shape reconstruction literature, Chamfer distance (CD) and Earth Mover's distance (EMD) were used to evaluate each method. Unlike CD, EMD has a number of different implementations for solving a sub-optimisation problem about the transportation of mass. In this study, we borrowed the EMD implementation code from Liu et al. (2020). In the experiment, we fixed the number of NTK1 network layers as 1. This is because there is no significant performance change when different number of network layers is used. The shape reconstruction using neural varifold is heavily influenced by the approaches from kernel interpolation (Cuomo et al., 2013) and neural splines (Williams et al., 2021). The implementation details are available at Appendix A.1. In addition, the shape reconstruction results with different number of points (i.e., 512 and 1024) are available at Appendix A.4.4. The visualisation of the ShapeNet reconstruction performance by all the methods compared is available at Appendix A.5.

Lastly, for point cloud based shape matching, simple 3-layer MLP neural networks were trained for deforming the source shape to the target shape with different shape similarity metric losses including neural varifold. The first example is deforming the source unit sphere into the target dolphin shape; the second is matching two different cup designs; the third is matching between two hippocampi; and more results are available at Appendix A.3. The data is acquired from the PyTorch3D, SRNFmatch and KeOps GitHub repositories (Ravi et al., 2020; Martin Bauer & Hsieh, 2020; Charlier et al., 2021). This experiment evaluates how well the source shape can be deformed based on the chosen shape similarity measure as the loss function. A simple 3-layer MLP network was solely trained with a single shape similarity measure loss, with the learning rate fixed to 1E-3 and the Adam optimizer. The network was trained with popular shape similarity measures including the CD, EMD, Charon-Trouvé varifold norm, and the proposed neural varifold norms (NTK1 and NTK2). In the case of CD and EMD, we followed the same method used for shape reconstruction. For varifold metrics, we used Equation 15; note that it is a squared distance commonly used for optimisation. For the numerical evaluation as a metric in Table 3, the square-root of Equation 15 was used. To be consistent with shape classification experiments, we chose the 5-layer NTK1 and

9-layer NTK2 to train and evaluate the similarity between two shapes. The detailed analysis for the role of the neural network layers on shape matching is available at Appendix A.4.3. The final outputs from the networks were evaluated with all of the shape similarity measures used in the experiments.

## 4.1 SHAPE CLASSIFICATION WITH LIMITED DATA

Small-data tasks are prevalent in applications when data is limited/scarce. Below we compare the performance of the finite-width neural networks and kernel methods when few training samples are given (Arora et al., 2019b). Table 1 presents the ModelNet classification accuracy with limited number of training samples. In general, the kernel based approaches show their strength for those small-data tasks. In detail, if only one training sample is used, then all

Table 1: ModelNet classification with limited training samples selected randomly. Every value indicates the average classification accuracy with standard deviation from 20 times iterations.

| Methods | 1-sample | 5-sample | 10-sample | 50-sample |
|---|---|---|---|---|
| | ModelNet10 | | | |
| PointNet | $38.84 \pm 6.41$ | $76.57 \pm 2.28$ | $84.14 \pm 1.43$ | $91.42 \pm 0.89$ |
| DGCNN | $33.56 \pm 4.60$ | $75.81 \pm 2.40$ | $83.90 \pm 1.70$ | $\mathbf{91.54 \pm 0.68}$ |
| Charon-Trouvé | $59.06 \pm 4.76$ | $78.64 \pm 2.90$ | $83.35 \pm 1.57$ | $87.98 \pm 0.79$ |
| NTK1 | $59.49 \pm 4.80$ | $81.34 \pm 2.78$ | $86.07 \pm 1.62$ | $90.18 \pm 0.93$ |
| NTK2 | $\mathbf{59.64 \pm 5.50}$ | $\mathbf{81.74 \pm 3.15}$ | $\mathbf{86.12 \pm 1.56}$ | $90.10 \pm 0.73$ |
| | ModelNet40 | | | |
| PointNet | $33.11 \pm 3.28$ | $63.30 \pm 2.12$ | $73.63 \pm 1.06$ | $85.43 \pm 0.31$ |
| DGCNN | $36.04 \pm 3.22$ | $67.49 \pm 1.80$ | $\mathbf{77.04 \pm 0.81}$ | $\mathbf{88.17 \pm 0.57}$ |
| Charon-Trouvé | $37.71 \pm 3.42$ | $60.43 \pm 1.51$ | $67.13 \pm 1.11$ | $77.20 \pm 0.54$ |
| NTK1 | $\mathbf{44.03 \pm 3.51}$ | $\mathbf{69.30 \pm 1.48}$ | $75.81 \pm 1.23$ | $83.88 \pm 0.53$ |
| NTK2 | $42.85 \pm 3.51$ | $67.81 \pm 1.47$ | $74.62 \pm 1.00$ | $83.26 \pm 0.42$ |

kernel based methods reveal their dramatically strong performance in both ModelNet10 and ModelNet40 classification in comparison to the finite-width neural networks like PointNet and DGCNN, with NTK2 and NTK1 achieving the best classification results on ModelNet10 and ModelNet40 classification tasks, respectively. Interestingly, if only a single sample is used, the performance of the Charon-Trouvé kernel is as good as the neural varifold approaches (NTK1 and NTK2) on ModelNet10 classification; however, its performance significantly drops on the ModelNet40 classification task. Analogous results are obtained when five samples are used for training. NTK1 and NTK2 achieve similar results (i.e., 81.3% and 81.7%) on ModelNet10, while Charon-Trouvé, PointNet and DGCNN underperformed by 3.1%, 5.1% and 5.9%, respectively; in the case of ModelNet40, NTK1 outperforms all other methods with higher gain comparing to the results on ModelNet10. As the number of training samples increases, the finite-width neural network based approaches significantly improve their performance on both ModelNet10 and ModelNet40 classification tasks. When ten samples are used for training, the proposed NTK1 and NTK2 show around 86.1% accuracy in Table 1, outperforming other methods with a small margin (i.e., 2∼3%) on ModelNet10, while DGCNN can outperform NTK as well as PointNet on ModelNet40. When 50 sample are used for training, then both PointNet and DGCNN outperform the NTK approaches with around 1% margin on ModelNet10 and 3∼5% margin on ModelNet40. Overall, NTK1 and NTK2 show similar performance (i.e., 0.3% difference) on ModelNet10, while NTK1 performs slightly better than NTK2 on ModelNet40 by 0.6∼1.6%. It is worth highlighting that our proposed NTK1 and NTK2 outperform the Charon-Trouvé varifold kernel in all the cases.

In terms of the computational efficiency, it is well known that kernel based learning has quadratic computational complexity. Interestingly, our proposed NTK1 and NTK2 are computationally competitive in the limited data scenario. For example, the computational cost for 5-sample training on ModelNet10 for NTK1 and NTK2 is respectively 47 and 18 seconds, while it takes respectively 254 and 502 seconds for training PointNet and DGCNN epochs with a single 3090 GPU.

## 4.2 SHAPE RECONSTRUCTION

Shape reconstruction from point clouds is tested for NTK1 as well as the state-of-the-art methods SIREN, neural splines and NKSR. Note that NTK2 is excluded in this test as it is not suitable to reconstruct the given shapes from point clouds. To be consistent with the existing shape reconstruction ways from point clouds, the quality of the reconstruction is evaluated with two popular shape similarity metrics – CD and EMD. Figure 1 showcases some shape reconstruction examples (e.g., airplane and cabinet) of the methods compared, with 2048 points samples. The performance of our NTK1 is visually better in terms of the surface completion and smoothness.

Quantitatively, Table 2 shows the mean and median of using the CD and EMD for 20 shapes randomly selected from each of the 13 different shape categories in the ShapeNet dataset. For the CD,

Table 2: ShapeNet 3D mesh reconstruction with 2048 points (mean/median values ×1E3).

| Metric | Method | Airplane | Bench | Cabinet | Car | Chair | Display | Lamp | Speaker | Rifle | Sofa | Table | Phone | Vessel |
|---|---|---|---|---|---|---|---|---|---|---|---|---|---|---|
| CD (mean) | SIREN | 1.501 | 1.624 | 2.430 | 2.725 | **1.556** | **2.193** | 1.392 | 7.906 | 1.212 | 1.734 | 1.856 | **1.478** | 2.557 |
|  | Neural Splines | 4.145 | **1.304** | 1.969 | 2.131 | 1.828 | 4.577 | **1.062** | **2.798** | **0.400** | **1.650** | **1.576** | 10.058 | 2.210 |
|  | NKSR | 1.141 | 2.000 | 2.423 | 2.198 | 2.520 | 17.720 | 5.477 | 3.622 | 0.414 | 1.848 | 2.493 | 1.547 | 1.093 |
|  | NTK1 | **0.644** | 1.314 | **1.991** | **2.107** | 1.734 | 4.666 | 1.134 | 2.806 | 0.425 | 1.654 | 1.586 | 10.397 | **1.079** |
| CD (median) | SIREN | 0.733 | 1.384 | 2.153 | 2.134 | 1.230 | 1.469 | 0.661 | 3.304 | 0.581 | 1.706 | 1.670 | 1.424 | 1.112 |
|  | Neural Splines | 0.947 | 1.289 | 1.799 | **1.640** | **1.160** | **1.413** | **0.479** | **2.749** | 0.347 | 1.586 | 1.372 | 1.600 | **0.788** |
|  | NKSR | 1.205 | 1.426 | **1.797** | 1.830 | 1.236 | 1.565 | 1.579 | 2.945 | **0.326** | 1.638 | 1.637 | **1.305** | 0.894 |
|  | NTK1 | **0.621** | **1.259** | 1.828 | 1.836 | 1.237 | 1.499 | 0.566 | 2.794 | 0.352 | **1.578** | **1.350** | 1.558 | 0.797 |
| EMD (mean) | SIREN | **2.990** | 3.763 | 4.983 | 5.208 | **4.649** | **4.658** | 24.068 | 13.292 | 2.418 | **3.688** | 8.745 | **3.237** | 4.500 |
|  | Neural Splines | 22.004 | **3.571** | **4.420** | **4.694** | 7.916 | 9.205 | **16.786** | **5.857** | **1.503** | 3.706 | **4.194** | 17.846 | 5.957 |
|  | NKSR | 7.153 | 8.456 | 8.018 | 8.190 | 16.824 | 31.182 | 21.182 | 9.984 | 2.329 | 5.871 | 13.658 | 4.152 | 4.581 |
|  | NTK1 | 3.120 | 4.153 | **4.420** | 4.767 | 7.350 | 9.653 | 23.381 | 6.236 | 1.592 | 3.888 | 5.259 | 24.101 | **3.534** |
| EMD (median) | SIREN | **2.690** | 2.938 | 4.520 | **3.803** | **4.411** | **3.314** | 6.240 | 6.240 | 1.605 | 3.653 | 3.782 | **3.060** | 2.576 |
|  | Neural Splines | 6.873 | **3.068** | **4.154** | 3.999 | 4.740 | 4.053 | 3.802 | **5.123** | **1.216** | **3.543** | **3.695** | 3.838 | 2.210 |
|  | NKSR | 5.732 | 5.119 | 4.440 | 5.313 | 5.683 | 3.777 | 4.927 | 5.975 | 1.227 | 3.641 | 6.375 | 3.088 | 2.771 |
|  | NTK1 | 2.864 | 3.319 | 4.284 | 3.947 | 5.293 | 3.875 | 3.288 | 5.795 | 1.271 | 3.738 | 3.980 | 3.380 | **2.074** |

NTK1 shows the best average reconstruction results for the airplane, cabinet, car and vessel categories, SIREN shows the best reconstruction results for the chair, display and phone categories; and the neural splines method shows the best reconstruction results for the rest 6 categories. NTK1 based reconstruction achieves the lowest mean EMD for vessel and cabinet, while neural splines and SIREN achieve the lowest mean EMD for 7 and 5 categories, respectively. NKSR does not achieve the lowest mean CD and EMD for all the categories.

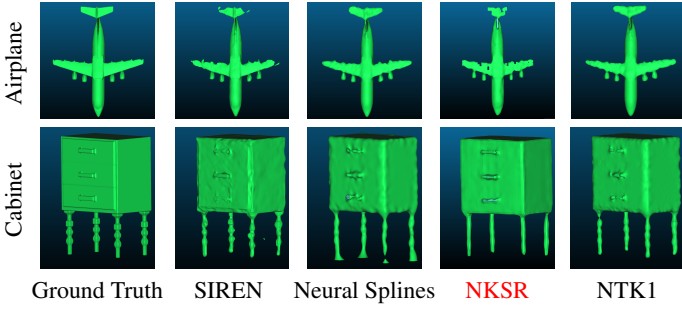

Ground Truth SIREN Neural Splines NKSR NTK1

Figure 1: Examples of shape reconstruction.

SIREN shows the lowest distance for both CD and EMD followed by NTK1. Surprisingly, the neural splines method underperforms in both the CD and EMD when we consider all the 13 categories. The performance of NTK1 on shape reconstruction is clearly comparable with these state-of-the-art methods. This might be counter-intuitive as it regularises the kernel with additional normal information. As shown in Appendix A.1, implicit neural representations with kernel ridge regression already incorporate the normal information by the definition of $\hat{\mathcal{X}} = \mathcal{X} \cup \mathcal{X}_\delta^- \cup \mathcal{X}_\delta^+$ and $\hat{\mathcal{Y}} = \mathcal{Y} \cup \mathcal{Y}_\delta^- \cup \mathcal{Y}_\delta^+$. Furthermore, there is no straightforward way to assign normals on the regular grid coordinates, where the signed distance values are estimated by the kernel regression. Any arbitrary unit normal vectors can be used for computing the varifold norm on the regular grid, which may cause errors on the signed distance estimation.

## 4.3 SHAPE MATCHING

For shape matching, we test various shape similarity metrics as loss functions to deform the given source shape into the target shape. Figure 2 shows three examples of shape matching based on various shape similarity metric losses. The neural network trained with the CD can capture the geometric details in all examples; however, it often shows non-smooth and broken surface geometry. For the case of shape matching between the two hippocampi, CD tends to oversmooth the sharp edges on the top right side of the target hippocampus. The network trained with EMD worked well on the dolphin shape but it could neither capture geometric details nor surface consistency of the target cup or hippocampus shapes. This is probably because the parameter defined for solving the transportation plan is not sufficient enough to accurately match the shapes. Moreover, more iterations and lower convergence threshold in computing the transportation plan will make the network training highly inefficient. The network trained with neural varifold metrics (NTK1 and NTK2) heavily penalises the broken meshes and noises on the surface; therefore, it shows significantly better mesh quality for all examples. In the case of dolphin, few high frequency features are over-smoothed for the shape matching networks trained with NTK2, while NTK1 does not have this effect and achieves the good shape matching result. The shape matching network trained with Charon-Trouvé shows acceptable

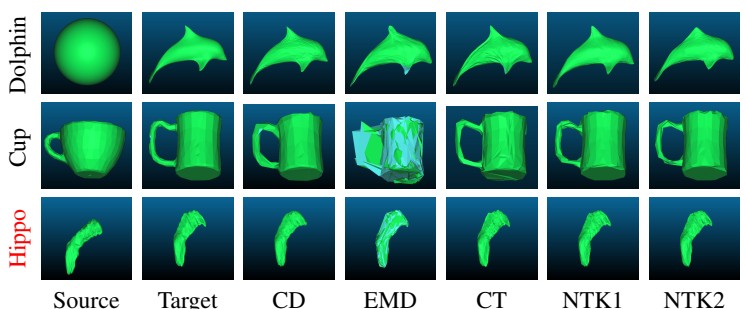

Figure 2: Shape matching examples with different shape similarity metrics. In particular, CT: Charon-Trouvé. Hippo is a shortened term referring to the hippocampus.

results for all examples. Note that one significant limitation of the Charon-Trouvé method is that the radial basis kernel used in it is highly sensitive to the point clouds density. It is therefore necessary to adjust the hyperparameter $\sigma$ of the radial basis kernel for each pair of point clouds (the parameter $\sigma$ is set to 0.05, 0.2 and 0.05 for the dolphin, cup and hippocampus shapes, respectively); otherwise, the Charon-Trouvé method may not converge or just lead to poor shape matching results.

Table 3 presents the quantitative evaluation of the shape matching task. Each column indicates that the shape matching neural network is trained with a specific shape similarity metric as the loss function. In the case of dolphin, when the evaluation metric is the same as the loss function used to train the network, then the network trained with the same evaluation metric achieves the best results. This is natural as the neural network is trained to minimise the loss function. It is worth highlighting that the shape matching network trained with the NTK1 loss achieves the second best score for all evaluation metrics except for itself. In other words, NTK1 can capture common characteristics of all shape similarity metrics used to train the network. Fur-

Table 3: Results of shape matching deforming the given source shapes into the target shapes using a neural network training with various shape similarity metrics. Metrics used in columns and rows are to train the neural network and for quantitative evaluation, respectively. Every value indicates the shape matching distance. In particular, the lowest one in each row (i.e., the best) is highlighted.

| | Metric | CD | EMD | CT | NTK1 | NTK2 |
|---|---|---|---|---|---|---|
| Dolphin | CD | **2.49E-4** | 3.39E-4 | 2.90E-4 | 2.84E-4 | 3.04E-4 |
| | EMD | 7.56E0 | **3.87E0** | 4.15E0 | 4.13E0 | 4.27E0 |
| | CT | 3.76E-2 | 2.94E-2 | **1.22E-2** | 1.63E-2 | 1.95E-2 |
| | NTK1 | 6.56E-3 | 1.89E-3 | 2.93E-3 | **4.82E-4** | 6.34E-4 |
| | NTK2 | 1.72E-2 | 4.33E-3 | 9.99E-3 | 1.34E-3 | **1.25E-3** |
| Cup | CD | 4.55E-3 | 9.74E-3 | 4.13E-3 | **3.26E-3** | 3.36E-3 |
| | EMD | 2.03E1 | 3.53E1 | 2.06E1 | 1.85E1 | **1.79E1** |
| | CT | 6.90E-1 | 2.85E0 | 4.07E-1 | 3.29E-1 | **3.20E-1** |
| | NTK1 | 1.72E-2 | 7.27E-1 | 1.97E-2 | **6.07E-3** | 6.50E-3 |
| | NTK2 | 3.14E-2 | 3.29E0 | 4.53E-2 | 1.34E-2 | **1.21E-2** |
| Hippocampus | CD | 3.49E-1 | 3.2E-1 | **2.43E-1** | 2.67E-1 | 2.65-1 |
| | EMD | 2.80E5 | 2.10E5 | 2.25E5 | 2.09E5 | **1.96E5** |
| | CT | 2.27E3 | 2.92E5 | 2.32E3 | 2.19E3 | **2.15E3** |
| | NTK1 | 1.84E5 | 1.01E9 | 59.7E5 | **4.93E3** | 9.98E3 |
| | NTK2 | 6.37E4 | 3.09E9 | 1.56E6 | **1.54E3** | **1.54E3** |

thermore, in the case of shape matching between two different cups, our neural varifold metrics (NTK1 and NTK2) achieve either the best or second best results regardless which shape evaluation metric is used. This indicates that the neural varifold metrics can capture better geometric details as well as surface smoothness for the cup shape than other metrics. In the case of shape matching between the source hippocampus and the target hippocampus, the network trained with CT excels in the CD metric, while the network trained with NTK1 achieves superior results with respect to NTK1 and NTK2 metrics. The shape matching network trained with NTK2 outperforms in EMD, CT and NTK2 metrics.

## 5 CONCLUSION

This paper presented the neural varifold as a highly competitive alternative representation to quantify geometry of point clouds for various applications including shape classification, reconstruction and matching. Detailed evaluation and comparison to the state-of-the-art methods demonstrate that the proposed versatile neural varifold is superior in shape classification particularly in the data scarcity scenario and is quite competitive for shape reconstruction and matching. In the future, to further enhance the representation performance in surface geometry, new network designs and their corresponding neural tangent kernels are of great interest to explore.

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

# A APPENDIX

## A.1 KERNEL BASED SHAPE RECONSTRUCTION

Given a set of surface points $\mathcal{X} = \{x_1, \ldots, x_k\}$ and the corresponding normals $\mathcal{N} = \{n_1, \ldots, n_k\}$ from an unknown surface $\mathcal{M}$, i.e., $\mathcal{X} \subset \mathcal{M}$ by definition of the implicit surface representation. All the surface points satisfy the property $f(x) = 0, \forall x \in \mathcal{M}$ for an unknown function $f$. The best way to approximate the function $f$ is to generate off-surface points and interpolate zero iso-surface. Given $\mathcal{Y} = \{y_1, \ldots, y_k\}, \forall y_i = 0$ and the distance parameter $\delta$, we define $\mathcal{X}_\delta^- = \{x_1 - \delta n_1, \cdots, x_k - \delta n_k\}$, $\mathcal{X}_\delta^+ = \{x_1 + \delta n_1, \cdots, x_k + \delta n_k\}$, $\mathcal{Y}_\delta^- = \{-\delta, \cdots, -\delta\}$, and $\mathcal{Y}_\delta^+ = \{\delta, \ldots, \delta\}$ in a similar manner. With union of the sets $\hat{\mathcal{X}} = \mathcal{X} \cup \mathcal{X}_\delta^- \cup \mathcal{X}_\delta^+$ and $\hat{\mathcal{Y}} = \mathcal{Y} \cup \mathcal{Y}_\delta^- \cup \mathcal{Y}_\delta^+$, the training data tuple $(\hat{\mathcal{X}}, \hat{\mathcal{Y}})$ can be used to represent the *implicit representation of surface geometry*.

Let us define regular voxel grids $\mathcal{X}_{\text{test}}$ on which all the extended point clouds $\hat{\mathcal{X}}$ lie. Note that there is no straightforward way to define normal vectors on the regular voxel grids, which are required for PointNet-NTK1 computation. Here, we assign their normals as the unit normal vector to z-axis. Then the signed distance corresponding to the regular grid $\mathcal{X}_{\text{test}}$ can be computed by kernel regression with neural splines or PointNet-NTK1 kernels $K(\mathcal{X}_{\text{train}}, \mathcal{X}_{\text{train}})$ and $K(\mathcal{X}_{\text{test}}, \mathcal{X}_{\text{train}})$, i.e.,

$$\mathcal{Y}_{\text{test}} = K(\mathcal{X}_{\text{test}}, \mathcal{X}_{\text{train}})[K(\mathcal{X}_{\text{train}}, \mathcal{X}_{\text{train}}) + \lambda I]^{-1} \mathcal{Y}_{\text{train}}, \tag{17}$$

where $\mathcal{Y}_{\text{train}}$ and $\mathcal{Y}_{\text{test}}$ are the signed distances for the extended point clouds and the regular grids, respectively. With the marching cube algorithm in Lorensen & Cline (1998), the implicit signed distance values on the regular grid with any resolution can be reformulated to the mesh representation.

## A.2 SHAPE CLASSIFICATION WITH THE FULL MODELNET DATASET

The overall shape classification accuracy with neural varifold and the comparison with state-of-the-art methods on both ModelNet10 and ModelNet40 are given in Table 4, where the entire training data is used. It shows that the finite-width neural network based shape classification methods (i.e., PointNet, PointNet++ and DGCNN) in general outperform the kernel based approaches, i.e., Charon-Trouvé, NTK1 and NTK2. DGCNN shows the best accuracy on both ModelNet10 and ModelNet40 amongst the methods compared. In the case of kernel based methods, NTK1 outperforms both NTK2 and Charon-Trouvé. The results are largely expected since the infinite-width neural networks with either NTK or NNGP kernel representations underperform in comparison with the equivalent finite-width neural networks (Lee et al., 2020) when sufficient training sampes are available. The computational complexity of kernel-based approaches is quadratic. With the ModelNet10 dataset containing 4899 samples, NTK1 and NTK2 respectively require approximately 12 hours and 6 hours of training time, whereas PointNet and DGCNN achieve similar accuracy with nearly 1 hour of training time using the entire dataset.

Table 4: ModelNet classification.

| Methods | ModelNet10 | ModelNet40 |
|---|---|---|
| PointNet*[1] | 94.4 | 90.5 |
| PointNet++[2] | 94.1 | 91.9 |
| DGCNN [3] | **95.0** | **92.2** |
| Charon-Trouvé | 89.0 | 80.5 |
| **NTK1** | 92.2 | 87.4 |
| **NTK2** | 92.2 | 86.5 |

* Point cloud inputs are positions and unit normal vectors – 6-feature vectors; note that the original paper's reported accuracy for ModelNet40 is 89.2% with only positions forming 3-feature vectors as inputs.

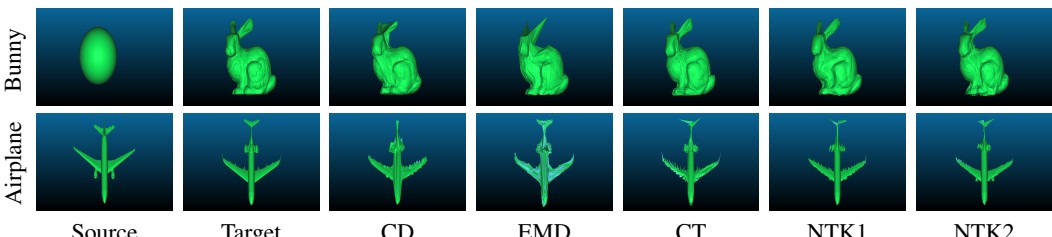

| | | | | | | |
|---|---|---|---|---|---|---|
| Source | Target | CD | EMD | CT | NTK1 | NTK2 |

Figure 3: Additional qualitative shape matching examples (i.e., Stanford bunny and airplane) with different shape similarity metrics. In particular, CT: Charon-Trouvé.

Table 5: Additional quantitative shape matching results (*cf.* Figure 3) on shapes Stanford bunny and airplane.

| | Metric | CD | EMD | CT | NTK1 | NTK2 |
|---|---|---|---|---|---|---|
| Bunny | CD | 9.32E-3 | 5.12E-3 | **3.60E-3** | 4.40E-3 | 4.32E-3 |
| | EMD | 2.31E4 | 4.74E3 | 3.72E3 | **3.13E3** | 3.52E3 |
| | CT | 2.40E-1 | 1.25E0 | **7.51E-2** | 1.28E-1 | 1.23E-1 |
| | NTK1 | 2.57E-2 | 1.32E-2 | 1.83E-3 | **2.22E-4** | 2.94E-4 |
| | NTK2 | 3.85E-2 | 2.68E-2 | 3.33E-3 | 8.85E-4 | **6.43E-4** |
| Airplane | CD | 1.36E-3 | 4.07E-4 | **3.72 E-3** | 3.81E-3 | 5.90E-3 |
| | EMD | 1.16E4 | 4.12E2 | **3.38E2** | 3.43E2 | 7.50E2 |
| | CT | 8.71E-2 | 3.62E0 | **-3.58E-4** | 1.67E-3 | 3.68E-3 |
| | NTK1 | 2.27E-3 | 1.80E-1 | 0.41E-6 | **0.31E-6** | 0.72E-6 |
| | NTK2 | 6.14E-2 | 5.13E0 | 8.69E-6 | 3.17E-6 | **2.42E-6** |

## A.3 Additional experimental results for shape matching

This section presents additional experiments for shape matching on two different shapes, i.e., Stanford bunny and airplane, see Figure 3 and Table 5 for the qualitative and quantitative results, respectively. For the assessment of shape matching between a sphere and the Stanford bunny, CT, NTK1 and NTK2 reveal detailed structure matching including ears, legs and tail. It is noteworthy that NTK1 tends to oversmooth the Stanford Bunny's face, whereas CT and NTK2 exhibit closer alignment to the ground truth. When matching the two airplanes, all methods fall short of achieving great matches, presenting broken meshes with incomplete details. For example, all methods fail to successfully match the tail wing and rear engines. In contrast, NTK1 and NTK2 showcase superior results concerning wings and fuselage, while CD, EMD and CT either exhibit heavily broken meshes or excessively smooth wings and fuselage.

Metrics used in columns and rows in Table 5 are to train the neural network and for quantitative evaluation, respectively. Every value indicates the shape matching distance. In particular, the lowest one in each row (i.e., the best) is highlighted. Quantitative results in Table 5 demonstrate that varifold metrics, including CT, NTK1 and NTK2, consistently outperform CD and EMD metrics. When shape matching between a sphere and the Stanford bunny, CT attains the best results in CD and CT metrics, NTK1 excels in EMD and NTK1 metrics, and NTK2 leads in its own metric NTK2. Interestingly, the shape matching results for the two airplanes, as depicted in Figure 3, indicate that CT, despite visual shortcomings, achieves the best scores in CD, EMD and CT metrics. NTK1 and NTK2 outperform others in their respective metrics. Notably, CT shows negative distance, primarily attributed to the point density parameter $\sigma$ in the CT metric causing instability.

Table 6: Shape classification performance of PointNet-NTK1 and PointNet-NTK2 with different number of neural network layers adopted in MLP and Conv1D on ModelNet40.

| Number of Layers | PointNet-NTK1 (5-sample) | PointNet-NTK2 (5-sample) |
|---|---|---|
| 1-layer MLP | $67.70 \pm 1.66$ | $64.70 \pm 1.34$ |
| 3-layer MLP | $69.06 \pm 1.57$ | $66.79 \pm 1.50$ |
| 5-layer MLP | $\mathbf{69.29 \pm 1.48}$ | $67.34 \pm 1.45$ |
| 7-layer MLP | $\mathbf{69.29 \pm 1.43}$ | $67.64 \pm 1.47$ |
| 9-layer MLP | $69.21 \pm 1.48$ | $\mathbf{67.81 \pm 1.47}$ |
| 1-layer Conv1D | $66.06 \pm 1.71$ | $63.20 \pm 1.30$ |
| 3-layer Conv1D | $68.82 \pm 1.62$ | $66.88 \pm 1.52$ |
| 5-layer Conv1D | $\mathbf{69.09 \pm 1.51}$ | $67.42 \pm 1.45$ |
| 7-layer Conv1D | $68.87 \pm 1.53$ | $67.77 \pm 1.41$ |
| 9-layer Conv1D | $68.68 \pm 1.46$ | $\mathbf{67.89 \pm 1.47}$ |

## A.4 Ablation analysis

### A.4.1 Neural varifolds with different number of neural network layers

This section shows the shape classification results based on different number of neural network layers. In this experiment, we randomly choose 5 samples per class on the training set of ModelNet40

and evaluate on its validation set. As shown in Section 4, we iterate the experiments 20 times with different random seeds. The key concept of the PointNet (Qi et al., 2017a) is the permutation invariant convolution operations on point clouds. For example, MLP or Conv1D with 1 width convolution window is permutation invariance. In this experiment, we choose different number of either MLP or Conv1D layers, and check how it performs on the ModelNet40 dataset. As shown in Table 6, the classification accuracy of PointNet-NTK1 with Conv1D operation is lower in comparison with the ones with MLP layers. In particular, 5-layer and 7-layer MLPs show similar performance with the PointNet-NTK1 architecture, i.e., 69.29% classification accuracy. In order to reduce the computational cost, we recommend fixing the number of layers in PointNet-NTK1 to 5. In the case of PointNet-NTK2, its performance increases as more layers are being added for it with both MLP and Conv1D operations. Furthermore, PointNet-NTK2 with Conv1D operation shows slightly higher classification accuracy in comparison with the ones with MLP layers. The percentage of the performance improvement becomes lower as the number of layers increases. In particular, 9-layer MLP versus 7-layer MLP for PointNet-NTK2 only brings 0.2% improvement; therefore, it is computationally inefficient to increase the number of layers anymore. Although PointNet-NTK2 with 9-layer Conv1D achieves 0.08% higher accuracy than the one with 9-layer MLP, PointNet-NTK2 with 9-layer MLP rather than Conv1D is used for the rest of the experiments in order to make the architecture consistent with the PointNet-NTK1.

### A.4.2 SHAPE CLASSIFICATION WITH DIFFERENT NEURAL NETWORK WIDTH

In this section, we analyse how the neural network width can impact on shape classification using the 9-layer MLP-based PointNet-NTK2 by varying the width settings from 128, 512, 1024 and 2048 to infinite-width configurations. We trained the model on 5 randomly sampled point clouds per class from the ModelNet10 training set. The evaluation was carried out on the ModelNet10 validation set. This process was repeated five times with different random seeds, and the average shape classification accuracy was computed. Notably, PointNet-NTK1 was excluded from this experiment due to the absence of a finite-width neural network layer corresponding to the elementwise product between two neural tangent kernels of infinite-width neural networks. The results presented in Table 7 demonstrate that the analytical NTK (infinite-width NTK) outperforms the empirical NTK computed from the corresponding finite-width neural network with a fixed width size. Furthermore, computing empirical neural tangent kernels with respect to different length of parameters is known to be expensive as the empirical NTK is expressed as the outer product of the Jacobians of the output of the neural network with respect to the parameters. The details of the computational complexity and potential acceleration have been studied in Novak et al. (2022). However, if the finite-width neural networks are trained with the standard way instead of using empirical NTKs on a large dataset (e.g. CIFAR-10), then finite-wdith neural networks can outperform the neural tangent regime with performance significant margins (Lee et al., 2020; Arora et al., 2019a). In other words, there is still a large gap understanding regarding training dynamics between the finite-width neural networks and their empirical neural kernel representations.

Table 7: Shape classification performance of 9-layer PointNet-NTK2 with different neural network width.

| Width for each layer | PointNet-NTK2 (5-sample) |
|---|---|
| 128-width | $78.74 \pm 3.30$ |
| 512-width | $80.08 \pm 3.02$ |
| 1024-width | $79.97 \pm 3.24$ |
| 2048-width | $80.46 \pm 3.13$ |
| infinite-width | $\mathbf{81.74 \pm 3.16}$ |

### A.4.3 SHAPE MATCHING WITH DIFFERENT NUMBER OF NEURAL NETWORK LAYERS

In this section, the behavior of the NTK pseudo-metrics with respect to different number of layers is evaluated. Note that the neural network width is not considered in this scenario as all pseudo-metrics are computed analytically (i.e., infinite-width). In this study, simple shape matching networks were trained solely by NTK psuedo-metrics with different number of layers. Table 8 shows that the shape matching network trained with the 5-layer NTK1 metric achieved the best score with respect to

Table 8: Ablation analysis for shape matching with respect to different number of neural network layers within NTK psueo-metrics. The number inside of the brackets (·) indicates the number of layers used for computing the NTK pseudo-metrics.

| Metric | NTK1 (1) | NTK1 (5) | NTK1 (9) |
|--------|----------|----------|----------|
| CD     | 2.82E-1  | **2.67E-1** | 2.99E-1  |
| EMD    | 2.43E5   | **2.09E5**  | 2.46E5   |
| CT     | 2.19E3   | **2.17E3**  | **2.17E3** |
| NTK1   | 7.74E3   | 4.93E3   | **4.90E3** |
| NTK2   | 2.56E3   | **1.54E3**  | 1.92E3   |
| **Metric** | **NTK2 (1)** | **NTK2 (5)** | **NTK2 (9)** |
| CD     | **2.59E-1** | 2.61E-1  | 2.64E-1  |
| EMD    | 2.14E5   | 2.32E5   | **1.93E5** |
| CT     | **2.15E3**  | 2.17E3   | **2.15E3** |
| NTK1   | 9.57E3   | **8.70E3**  | 9.98E3   |
| NTK2   | **1.28E3**  | 1.41E3   | 1.53E3   |

Table 9: ShapeNet 3D mesh reconstruction with 1024 points (mean/median values $\times$1E3).

| Metric | Method | Airplane | Bench | Cabinet | Car | Chair | Display | Lamp | Speaker | Rifle | Sofa | Table | Phone | Vessel |
|--------|--------|----------|-------|---------|-----|-------|---------|------|---------|-------|------|-------|-------|--------|
| CD (mean) | SIREN | **0.936** | **1.499** | 3.134 | 5.363 | **2.492** | **3.635** | 2.536 | 4.109 | 2.134 | 3.660 | 2.264 | **1.674** | 1.339 |
| | Neural Splines | 11.640 | 1.905 | **2.264** | 2.440 | 2.983 | 4.770 | **1.418** | **3.437** | **0.439** | 1.924 | 3.936 | 9.026 | 2.255 |
| | NKSR | 1.898 | 3.506 | 6.224 | **2.286** | 3.584 | 46.997 | 9.229 | 4.138 | 0.665 | 2.029 | 3.213 | 2.243 | **1.285** |
| | PointNet-NTK1 | 1.584 | 1.742 | 2.274 | 2.494 | 2.655 | 5.337 | 1.465 | 3.947 | 0.456 | **1.870** | **2.029** | 12.138 | 1.341 |
| CD (median) | SIREN | **0.756** | **1.272** | 2.466 | 2.305 | **1.281** | **1.385** | 1.156 | 3.411 | 0.487 | 1.706 | **1.601** | 1.390 | 1.040 |
| | Neural Splines | 8.171 | 1.562 | 1.830 | 2.058 | 2.152 | 1.548 | **0.698** | 3.071 | **0.359** | **1.657** | 1.715 | 1.594 | **0.879** |
| | NKSR | 1.900 | 2.245 | **1.799** | 2.190 | 2.116 | 1.880 | 2.347 | 3.488 | 0.407 | 1.697 | 1.695 | **1.345** | 0.956 |
| | PointNet-NTK1 | 0.820 | 1.701 | 1.933 | **1.995** | 1.522 | 1.719 | 0.733 | **3.045** | 0.366 | 1.719 | 1.643 | 1.658 | 1.016 |
| EMD (mean) | SIREN | **2.183** | **3.679** | 6.385 | 10.712 | **5.932** | **7.527** | **12.850** | 8.714 | 3.164 | 7.633 | **4.992** | **3.645** | **3.265** |
| | Neural Splines | 60.566 | 6.540 | 5.338 | **5.380** | 15.935 | 8.882 | 22.745 | **6.457** | 1.878 | **4.335** | 11.733 | 18.019 | 6.367 |
| | NKSR | 12.939 | 11.990 | 16.684 | 7.571 | 21.706 | 44.190 | 32.236 | 12.486 | 3.613 | 4.930 | 14.917 | 6.609 | 6.715 |
| | PointNet-NTK1 | 6.704 | 5.984 | **5.301** | 5.907 | 14.868 | 11.507 | 29.595 | 8.070 | **1.773** | 4.596 | 11.606 | 24.903 | 3.841 |
| EMD (median) | SIREN | **1.982** | **3.211** | 5.232 | 4.699 | **5.678** | **3.567** | **2.916** | 5.548 | 1.351 | 3.804 | **3.122** | 3.415 | 2.552 |
| | Neural Splines | 35.458 | 4.713 | **4.745** | 4.779 | 11.570 | 3.915 | 5.719 | **4.575** | **1.334** | 3.650 | 5.041 | 4.828 | 2.276 |
| | NKSR | 11.317 | 6.933 | 5.035 | 5.432 | 9.807 | 8.597 | 7.871 | 8.397 | 1.765 | **3.524** | 8.140 | **3.400** | 2.354 |
| | PointNet-NTK1 | 3.716 | 4.659 | 5.050 | **4.598** | 7.613 | 4.062 | 9.168 | 5.456 | 1.364 | 4.105 | 4.257 | 4.710 | **2.209** |

CD, EMD, CT and NTK2 metrics, while the one with the 9-layer NTK1 metric achieved the best score with respect to CT and NTK1 metrics. This is in accordance with the ablation analysis for shape classification, where 5-layer NTK1 achieved the best classification accuracy in the Model-Net10 dataset. In comparison, NTK2 shows a mixed signal. The shape matching network trained with the 1-layer NTK2 metric achieved the best outcome with respect to Chamfer, CT and NTK2 metrics, while the one trained with the 9-layer NTK2 achieved the best results with respect to EMD and CT metrics. The network trained with 5-layer NTK2 showed the best result with respect to the NTK1 metric. This is not exactly in accordance with respect to shape classification with the NTK2 metric, where the shape classification accuracy improves as the number of layers increases. However, training a neural network always involves some non-deterministic nature; therefore, it is yet difficult to conclude whether the number of neural network layers is important for improving the shape matching quality or not.

### A.4.4 Shape reconstruction with different point cloud sizes

In this section, we compare shape reconstruction results with different point cloud sizes, i.e., 512, 1024 and 2048 points. As indicated in Tables 2, 9 and 10, PointNet-NTK1 and neural splines show that the quality of the reconstructions is degraded as the number of points decreases. For NKSR, its reconstruction quality becomes worse as the number of point clouds decreases for most categories, but few categories (i.e., cabinet and vessel) show the opposite trend. In the case of SIREN, the convergence of the SIREN network plays more important role for the shape reconstruction quality. For example, the shape reconstruction results by SIREN on the airplane category show that the shape reconstruction with 1024 points is better than that with 2048 points. This is due to the non-deterministic nature of DNN libraries, i.e., it is difficult to control the convergence of the SIREN network with our current experimental setting $10^4$ epochs. Note that the SIREN reconstruction is

Table 10: ShapeNet 3D mesh reconstruction with 512 points (mean/median values ×1E3).

| Metric | Method | Airplane | Bench | Cabinet | Car | Chair | Display | Lamp | Speaker | Rifle | Sofa | Table | Phone | Vessel |
|---|---|---|---|---|---|---|---|---|---|---|---|---|---|---|
| CD (mean) | SIREN | **1.385** | **1.992** | 14.975 | 4.323 | **2.813** | **3.094** | 7.874 | 5.426 | 3.731 | 3.582 | 10.423 | **2.524** | 2.278 |
| | Neural Splines | 21.410 | 3.752 | 2.818 | 2.985 | 5.217 | 5.089 | **2.050** | 4.393 | 0.565 | **2.228** | 5.953 | 8.721 | 2.699 |
| | NKSR | 3.974 | 6.265 | 3.545 | 2.594 | 5.348 | NA | 9.859 | 5.259 | 17.419 | 2.059 | 6.636 | 1.677 | 1.540 |
| | PointNet-NTK1 | 2.454 | 2.674 | **2.565** | 3.233 | 3.793 | 6.087 | 2.193 | **4.045** | **0.550** | 2.252 | **2.702** | 14.349 | **2.090** |
| CD (median) | SIREN | **0.715** | **1.678** | 3.635 | 3.122 | **1.914** | **1.672** | 1.540 | 4.707 | 1.156 | 2.256 | **1.746** | 1.497 | 1.130 |
| | Neural Splines | 21.040 | 2.466 | 1.935 | 2.369 | 3.347 | 2.058 | **1.023** | 3.361 | **0.385** | 1.918 | 2.411 | 1.717 | 1.226 |
| | NKSR | 2.627 | 3.336 | **1.894** | 2.015 | 3.752 | NA | 4.427 | 3.753 | 0.906 | 1.833 | 3.555 | **1.411** | **0.856** |
| | PointNet-NTK1 | 1.243 | 2.246 | 2.106 | 2.316 | 2.473 | 1.968 | 1.346 | **3.330** | 0.387 | 1.890 | 1.963 | 2.013 | 1.309 |
| EMD (mean) | SIREN | **3.411** | **5.833** | 24.404 | 9.460 | **7.366** | **6.558** | **26.828** | 13.584 | 5.224 | 6.457 | 16.578 | 4.764 | **4.831** |
| | Neural Splines | 120.415 | 11.749 | 7.478 | **6.057** | 26.382 | 11.486 | 30.216 | 8.686 | 3.048 | **5.128** | 25.433 | 19.087 | 8.431 |
| | NKSR | 24.959 | 21.190 | 11.433 | 9.346 | 30.485 | NA | 36.050 | 18.147 | 13.115 | 5.226 | 24.257 | **4.701** | 8.605 |
| | PointNet-NTK1 | 13.826 | 9.217 | **5.614** | 11.548 | 16.465 | 13.501 | 35.540 | **8.334** | **2.436** | 6.010 | **15.663** | 27.025 | 5.897 |
| EMD (median) | SIREN | **1.964** | **5.036** | 8.656 | 6.643 | **5.553** | **3.650** | 14.281 | 14.499 | 2.296 | 4.682 | **3.735** | 3.779 | 3.012 |
| | Neural Splines | 115.527 | 9.698 | 4.679 | **4.863** | 20.006 | 4.476 | 10.834 | **5.405** | **1.548** | 4.234 | 8.205 | 4.742 | 3.147 |
| | NKSR | 25.234 | 14.795 | **4.405** | 6.669 | 16.082 | NA | 10.727 | 8.655 | 3.132 | **4.147** | 9.839 | **3.595** | **2.650** |
| | PointNet-NTK1 | 9.863 | 6.122 | 4.758 | 7.171 | 6.822 | 5.076 | **9.296** | 5.683 | 1.626 | 4.497 | 7.455 | 6.658 | 3.313 |

NA indicates that the method fails to reconstruct few shapes in the given class.

computationally much more expensive (around 20∼30 minutes) than either the PointNet-NTK1, neural splines or the NKSR approach (around 1∼5 seconds).

## A.5 VISUALISATION OF SHAPENET RECONSTRUCTION RESULTS

In this section, we present additional visualisations of shape reconstruction outcomes obtained through three baseline methods (i.e., SIREN, neural splines, and NKSR), along with the proposed NTK1 method, across 13 categories of ShapeNet benchmarks. Five shape reconstruction results are illustrated for each category. Specifically, Figure 4 showcases examples from the Airplane, Bench, and Cabinet categories. Figure 5 exhibits five instances of shape reconstruction outcomes for the Car, Chair, and Display categories. Moving on to Figure 6, it displays examples from the Lamp, Speaker, and Rifle categories. Similarly, Figure 7 demonstrates five instances of shape reconstruction results for the Sofa, Table, and Phone categories. Finally, Figure 8 focuses on the shape reconstruction results for the Vessel category.

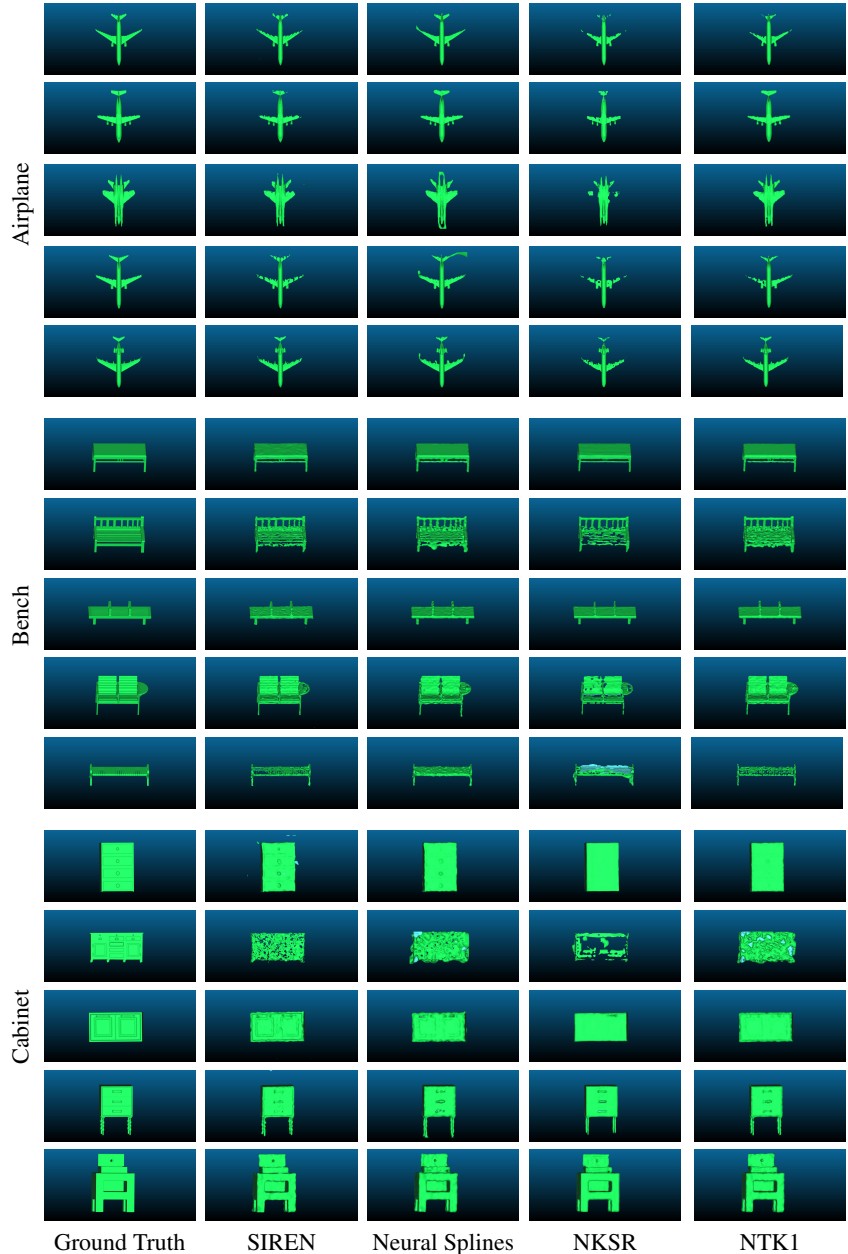

Figure 4: Visualisation of shape reconstruction results from SIREN, Neural Splines, NKSR and NTK1 for the Airplane, Bench and Cabinet categories.

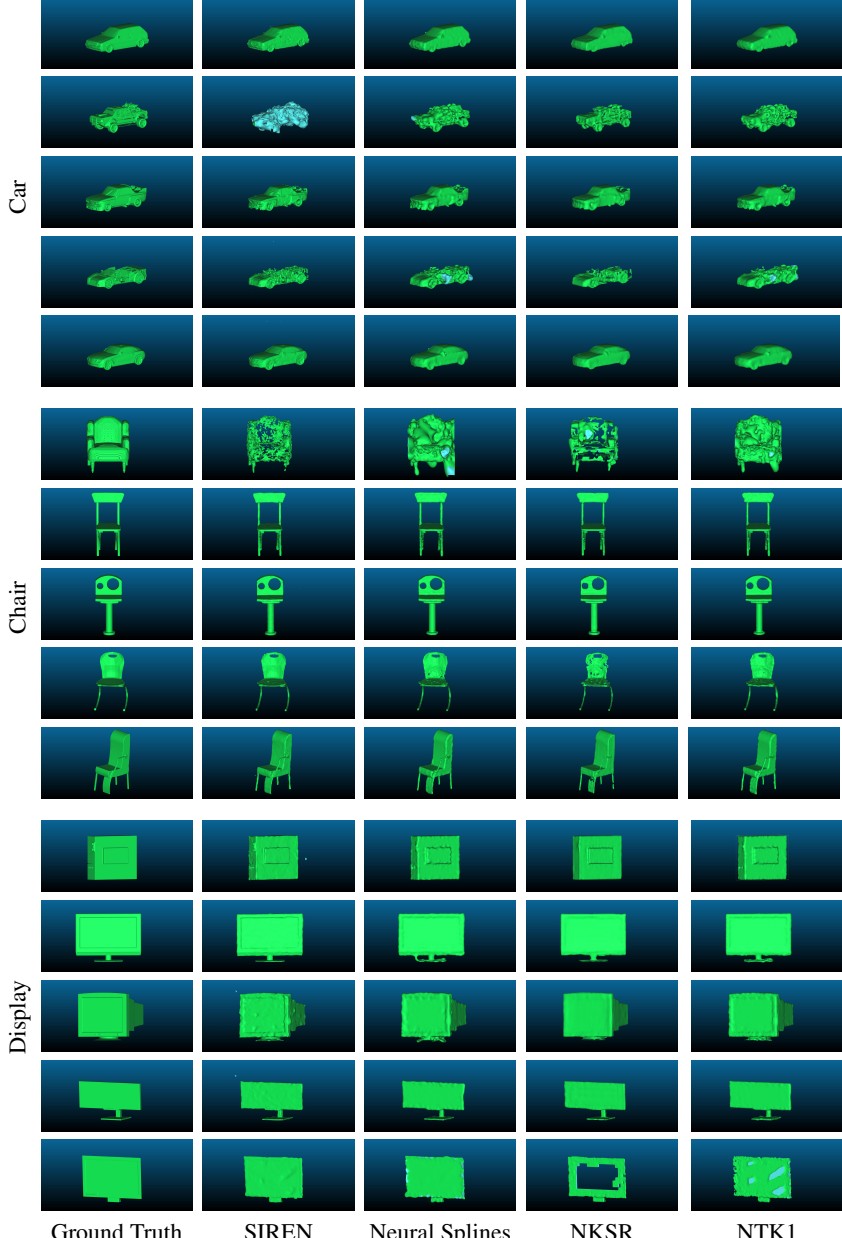

Figure 5: Visualisation of shape reconstruction results from SIREN, Neural Splines, NKSR and NTK1 for the Car, Chair and Display categories.

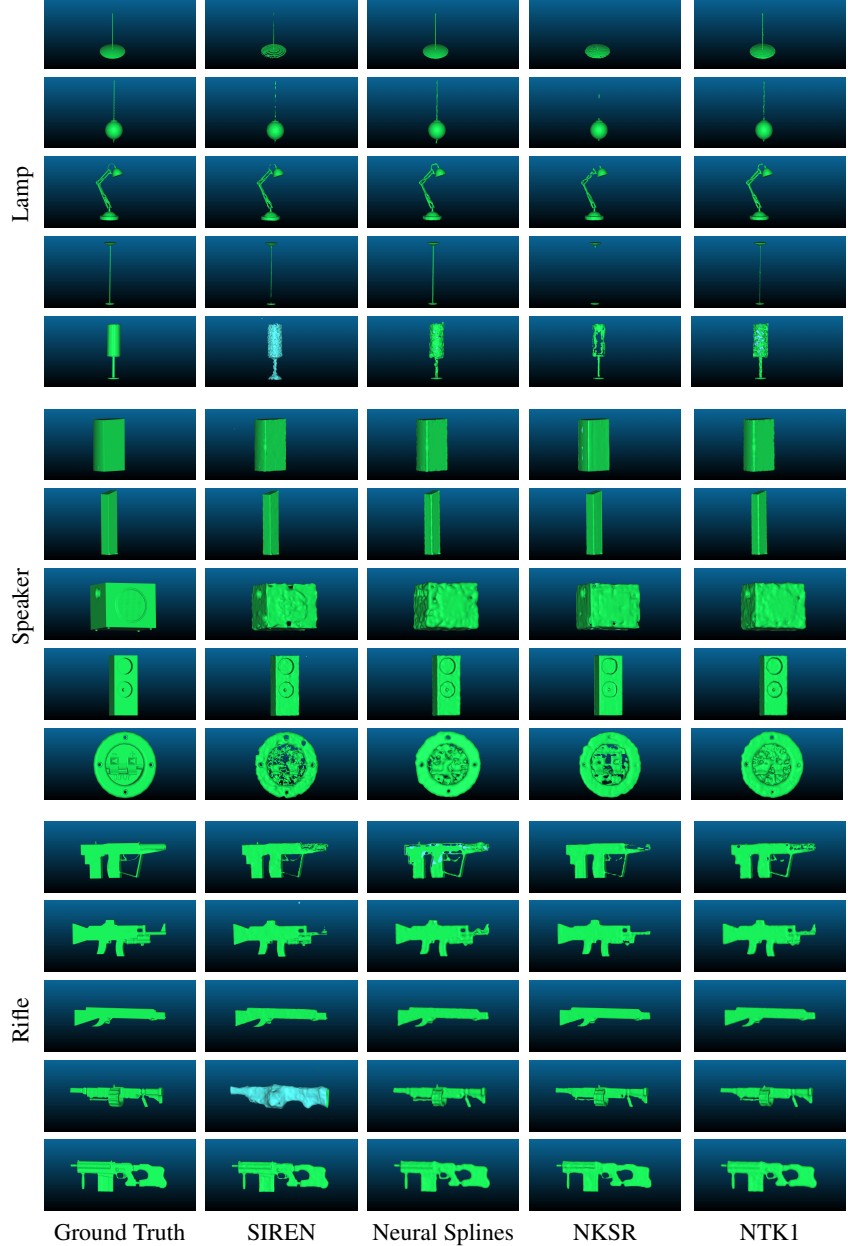

Figure 6: Visualisation of shape reconstruction results from SIREN, Neural Splines, NKSR and NTK1 for the Lamp, Speaker and Rifle categories.

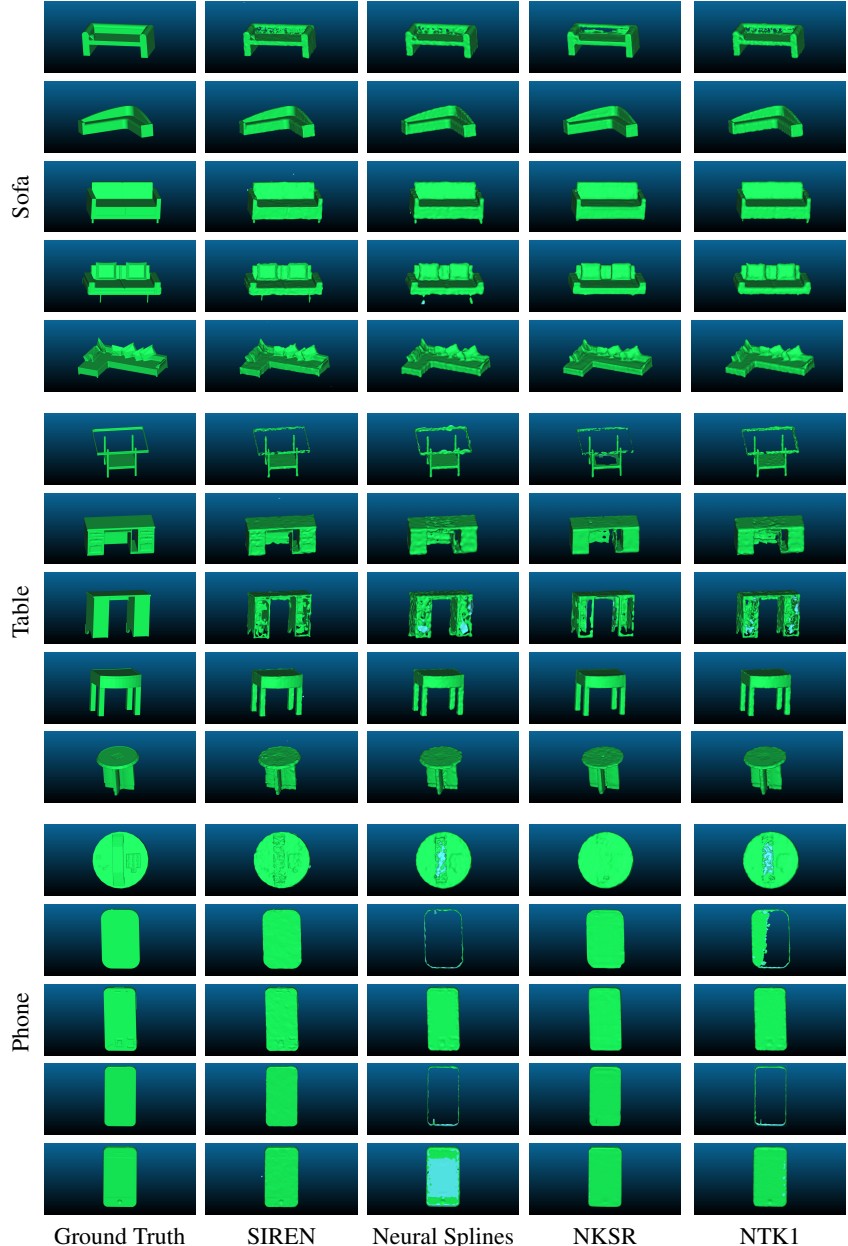

Figure 7: Visualisation of shape reconstruction results from SIREN, Neural Splines, NKSR and NTK1 for the Sofa, Table and Phone categories.

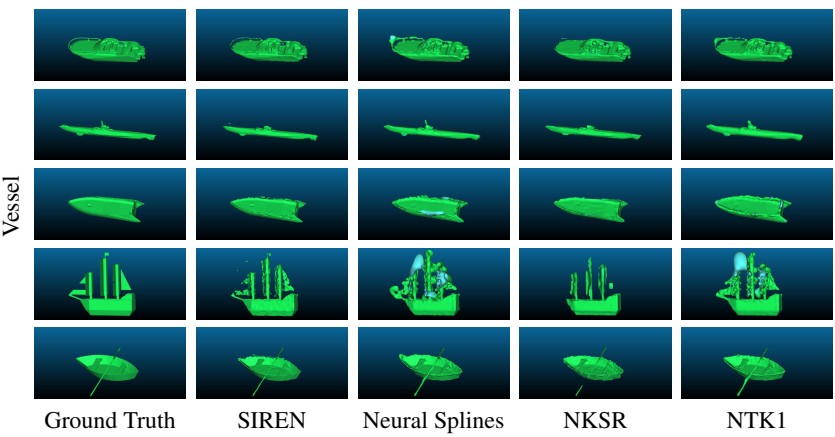

Vessel

Ground Truth    SIREN    Neural Splines    NKSR    NTK1

Figure 8: Visualisation of shape reconstruction results from SIREN, Neural Splines, NKSR and NTK1 for the Vessel category.

