# OpenReview forum: "Neural varifolds: an aggregate representation for quantifying geometry of point clouds"
_ICLR.cc/2024/Conference — Submitted to ICLR 2024_

### Official Review · Reviewer_vDG7 · 2023-10-30

**Soundness:** 2 fair
**Presentation:** 3 good
**Contribution:** 2 fair
**Rating:** 5
**Confidence:** 2

**Summary:**

The paper proposes neural varifold representations to characterize the geometry of point clouds. The neural varifolds combine point positions and tangent spaces to quantify surface geometry. Two algorithms are presented to compute neural varifold norms between point clouds using neural tangent kernels. The neural varifold is evaluated on shape classification, reconstruction, and matching tasks.

**Strengths:**

- The motivation for neural varifolds is well-articulated based on relevant literature from geometric measure theory and deep learning.
- The two proposed algorithms (PointNet-NTKI and PointNet-NTK2) to compute neural varifold are reasonable extensions of related work.
- Experiments are conducted on standard benchmarks to evaluate different tasks, with ablation studies and comparisons to baseline methods.

**Weaknesses:**

- The theoretical underpinnings of PointNet-NTK2 are less clear than PointNet-NTKI.
- The performance of neural varifolds is not state-of-the-art on most tasks.
- The network design in the manisript is relatively simple, which I am nore sure mainly results in the relatively poor performance. If so, could authors provide the design principle or experiments (if applicable) of the combination with more advanced networks to demonstrate the promising value of the proposed representation.
- More visualizations are favorble to highlight the characteristics and advantages of the proposed approach, which could be included in the supplementary file.

**Questions:**

- Can you elaborate more on the limitations of PointNet-NTK2 compared to PointNet-NTKI from a theoretical standpoint?
- How do you think the performance of neural varifolds can be further improved?
- What are possible directions to extend this work for future research?

---

> ### Author Response · Authors · 2023-11-21
> **Reply to Reviewer vDG7's Weakness 2**
>
> ##  • The performance of neural varifolds is not state-of-the-art on most tasks.
>
> We appreciate the reviewer's remark on the manuscript. To clear some misunderstanding regarding the computational complexity and methods' performance, we would also like to refer the reviewer to our reply to reviewer wzuY and reviewer 6p7s particularly for their first question. Please see below for our further response.
>
>
> Only for shape classification with all data, the performance of the neural varifold is not as comparable as the existing baselines like PointNet, DGCNN or more advanced methods, indicating the potential current limitation of the proposed methods in this specific case.  We would like to highlight that the performances of the proposed approaches on shape classification with  limited data, shape reconstruction and shape matching are either  better in many cases or at least comparable with other methods.
>
> In the limited data scenario, the proposed methods are actually not computationally heavier than neural networks with respect to the training time. For example, the computational time for 5-sample training on ModelNet10 is respectively 47 and 18 seconds for PointNet-NTK1 and PointNet-NTK2, while it takes 254 and 502 seconds for training PointNet and DGCNN with a single NVIDIA 3090 GPU, respectively. Although the inference time for the standard neural network could be cheaper after training completed, considering the performance gap and computational cost for our proposed methods, it is still within acceptable range and is worth improving further.
>
> In the case of shape reconstruction, Table 3 in the manuscript shows that none of the baseline methods are dominating the task and that the proposed method is competitive to the baseline methods. Note that we have also included a SOTA method -- neural kernel surface reconstruction (NKSR) -- into baseline in Table 3 in our revised manuscript. In addition, we have added more visualisation of the shape reconstruction results from all baseline methods as well as our proposed method.
>
> Our proposed method can also be used to train and evaluate the similarity between two shapes. As shown in the experiment of shape matching, our proposed method can achieve competitive results with respect to all baseline metrics including popular ones (e.g., Chamfer distance and Earth Mover's distance). Moreover, we have incorporated three additional shape matching results (i.e.,  1. two hippocampi, 2. sphere to Stanford Bunny, and 3. two airplanes) into the Appendix of the revised manuscript. In all three instances, both NTK1 and NTK2 exhibit excellent results; please also refer to our response to reviewer 6p7s's second question, in which new quantitative results are provided.

---

> ### Author Response · Authors · 2023-11-21
> **Reply to Reviewer vDG7's Weakness 3**
>
> ##  • The network design in the manuscript is relatively simple, which I am not sure mainly results in the relatively poor performance. If so, could authors provide the design principle or experiments (if applicable) of the combination with more advanced networks to demonstrate the promising value of the proposed representation.
>
> The current network designs (NTK1 and NTK2) are based on one of the most popular neural networks, PointNet, for 3D point clouds.
> They are heavily influenced by the PointNet architecture where the neural networks consist of a series of permutation invariant operations (e.g., MLP layers, Conv1D with 1-width convolution kernel, average pooling, etc.). Therefore, the performance of the neural network might be limited by the current design of the  PointNet architecture, where local variations are hard to capture.
>
> PointNet can be viewed as a graph convolution without edge connections (i.e., isolated nodes). Neural tangent kernels on graph convolution may improve the performance on shape classification and/or shape reconstruction [2]. However, computation and memory requirement of graph neural tangent computation are currently much more expensive than that of PointNet-like architectures. Although sparse representation of graphs as well as efficient graph parallelism can resolve the computation as well as memory requirement bottleneck, it is an active area of research. In our experience, computational cost for graph convolutions is too heavy, while the performance benefits at the expense of high computational cost are limited.
>
> We have tested a simple vanilla GNN architecture by adding an additional graph aggregation (message passing) layer on top of each layer of MLP on both PointNet-NTK1 and PointNet-NTK2. We refer these two algorithms as vanilla graph neural tangent kernel1 (vanilla-GNTK1) and  vanilla graph neural tangent kernel2 (vanilla-GNTK2), respectively.  The table below shows the performance comparison between the proposed PointNet NTKs and vanilla GNTKs. Note that PointNet-NTK1 and PointNet-NTK2 respectively take 47 and 18 seconds, while vanilla-GNTK1 and vanilla-GNTK2 respectively take 391 and 193 seconds on 5-sample ModelNet10 classification tasks. Given the computational time and performance comparisons, merely translating advanced architectures in a naive manner may not necessarily enhance performance. Consequently, further in-depth research is imperative to achieve performance improvement. We will leave it for the future work.
>
> | Model Architecture | ModelNet10 (5-sample) |
> |--------------------|-----------------------|
> | PointNet-NTK1      | 81.34 $\pm$ 2.78      |
> | Vanilla-GNTK1      | 81.91 $\pm$ 2.95      |
> | PointNet-NTK2      | 81.74 $\pm$ 3.16      |
> | Vanilla-GNTK2      | 81.67 $\pm$ 3.20      |

---

> ### Author Response · Authors · 2023-11-21
> **Reply to Reviewer vDG7's Weakness 4**
>
> ##  • More visualizations are favorble to highlight the characteristics and advantages of the proposed approach, whichcould be included in the supplementary file.
>
> Thanks for the reviewer's suggestion. More visualisation regarding shape matching as well as shape reconstruction will be added into the Appendix of the revised manuscript, which will be uploaded to OpenReview before the authors-reviewers discussion deadline.

---

> ### Author Response · Authors · 2023-11-21
> **Reply to Reviewer vDG7's Question 2 & 3**
>
> ##  • How do you think the performance of neural varifolds can be further improved?
>
> ##  • What are possible directions to extend this work for future research?
>
> Thanks for the reviewer's open questions. In order to minimise the performance gap, feature augmentation techniques can be explored. In the case of CIFAR10 classification, the work in [1] augmented the CIFAR dataset by a factor of 20, and then used highly parallelised conjugate gradient method to solve the linear system and achieved 91.2\% accuracy, which is the current SOTA amongst kernel based methods. Although its performance is not as good as the SOTA neural networks, it could be further improved in the context of few-shot learning.
>
> Our architecture is based on a simple PointNet architecture. Even though its performance is quite impressive already, it shares the drawback of the PointNet architecture and thus local variations cannot be able to be captured very efficiently as local neighbourhood is not considered in the architecture. It might be worth investigating graph neural tangent kernel and its variants [2] to take into account subtle variations within the local neighbourhood.  With the potential improvement of the computational complexity [3,4], it might be able to achieve better performance on few-shot learning, shape reconstruction and shape matching. Furthermore, various SOTA architectures like Point Transformer can be exploited for the improvement of the architecture side. This is of great interest to investigate in future study.
>
> From the varifold theory point of view, the multiplicity function $\theta$ in  Definition 1 can also be used to differentiate the local density or second-order variations of the point clouds. In the current study, the multiplicity $\theta$ is defined as constant 1 for simplicity, it might be worth investigating whether the multiplicity term can be learned by data-driven approaches like [5] to boost the performance. Furthermore, varifold framework is flexible such that it can take into account additional information on the surface. For example, Equation (32) in [6] defines the functional varifold, which can take into account texture information as a function. By doing so, the varifold framework can take advantage of the shape with texture information for various tasks, e.g. texture analysis, shape registration, shape reconstruction with texture, etc.  We believe these are just some of the many possible directions to extend this work for future research.

---

> ### Author Response · Authors · 2023-11-21
> **References**
>
> [1] Adlam, Ben, et al. "Kernel Regression with Infinite-Width Neural Networks on Millions of Examples." arXiv preprint arXiv:2303.05420 (2023).
>
> [2] Du, Simon S., et al. "Graph neural tangent kernel: Fusing graph neural networks with graph kernels." Advances in neural information processing systems 32 (2019).
>
> [3] Zandieh, Amir, et al. "Scaling neural tangent kernels via sketching and random features." Advances in Neural Information Processing Systems 34 (2021): 1062-1073.
>
> [4] Han, Insu, et al. "Fast neural kernel embeddings for general activations." Advances in neural information processing systems 35 (2022): 35657-35671.
>
> [5] Huang, Jiahui, et al. "Neural Kernel Surface Reconstruction." Proceedings of the IEEE/CVF Conference on Computer Vision and Pattern Recognition. 2023.
>
> [6] Charlier, Benjamin, Nicolas Charon, and Alain Trouvé. "The fshape framework for the variability analysis of functional shapes." Foundations of Computational Mathematics 17 (2017): 287-357.
>
> [7] Shankar, Vaishaal, et al. "Neural kernels without tangents." International conference on machine learning. PMLR, 2020.

---

> ### Author Response · Authors · 2023-11-23
> **Reply to Reviewer vDG7's Weakness 1 & Question 1**
>
> ## • The theoretical underpinnings of PointNet-NTK2 are less clear than PointNet-NTKI.
> ## • Can you elaborate more on the limitations of PointNet-NTK2 compared to PointNet-NTKI from a theoretical standpoint?
>
> We appreciate the reviewer's  remark on the manuscript. The paragraph following Equation (13) in the original manuscript about the theoretical underpinnings of PointNet-NTK2 was not clear so we rewrote it in the revised manuscript. Here is the new version: ***"[PointNet-NTK2] cannot be associated in the limit with a Charon-Trouv\'e type kernel, in contrast with PointNet-NTK1, but it remains theoretically well grounded because the explicit coupling of positions and normals is a key aspect of the theory of varifolds that provides strong theoretical guarantees (convergence, compactness, weak regularity, second-order information, etc.). Furthermore, PointNet-NTK2 falls into the category of neural networks proposed for point clouds that treat point positions and surface normals as 6-feature vectors, and thus PointNet-NTK2 is a natural extension of current neural networks practices for point clouds."*** We hope that this new version clarifies the theoretical underpinnings of PointNet-NTK2.

---

### Official Review · Reviewer_6p7s · 2023-10-30

**Soundness:** 3 good
**Presentation:** 3 good
**Contribution:** 2 fair
**Rating:** 5
**Confidence:** 4

**Summary:**

In this paper, the authors propose to consider a novel representation, varifolds for neural point cloud analysis. They first familiarizes the readers with the idea of varifolds consisting of a positional and Grassmannian component, which can be viewed as the joint representation consisting of points and normals. Then, the concept of neural tangent kernels is introduced to link the kernel theory and neural networks.  The NTK determines the distance between varifolds using neural networks. The authors then propose to represent point clouds using NTK to allow comparison in the varifold representation for conducting downstream tasks.  To demonstrate the effectiveness of the proposal, the authors conduct experiments on point cloud classification, surface reconstruction and shape matching. Especially in tasks where the data is limited, the results demonstrate that the proposed representation is able to capture the characteristics of the underlying shape, showing promise for further analysis.

**Strengths:**

- The authors carefully introduces the idea of varifolds and how it can be used to represent the point clouds consisting of points and normals. They extensively explain how metrics can be introduced in this representation space.

- The authors thoroughly explain how neural tangent kernel can be used to introduce kernels into the domain of neural networks and how point cloud represented as varifolds can be compared. The representation as well as the derivation is  theoretically solid.

- They propose two variants of the varifold representation using neural networks: NTK-1 and 2. The first separates the positional mass and the normal elements computes them separately, while the second jointly handles points and normals, as conventional point cloud neural network models do.

- The task of using different metrics to conduct non-rigid registration is very interesting. As the proposed metric performs well in both cases of dolphin and cup, it seems to be a promising metric to compare different point cloud data and its underlying shapes.

**Weaknesses:**

- Despite the very interesting theoretical approach of using varifolds, its practicality remains questionable. The interesting results can be found in small-sample shape classification tasks, where the proposed method and representation outperformed other conventional methods. However in most other tasks, the method had been outperformed by conventional methods with presumably more expensive computation. Despite the authors’ claim that the representation is able to extract both global and local shape similarities, the experiments demonstrate otherwise. It would have been better if the authors proposed a specific network structure that is able to take better advantage of the NTK representation. In order to compensate for the practical disadvantages, the authors could have introduced theoretical advantages over the conventional approaches. Such seems to be missing in the paper.

- The non-rigid registration results are very interesting, however, only two samples were provided for the experiments. It would be more convincing to apply the method on various shapes to analyze the tendencies of the proposal. Therefore, I believe the experiments are incomplete.

- Some ablation study had been conducted in the supplementary material by changing the number of layers of the target network. However, as the neural tangent kernel is derived from the study that links kernels to over-parameterized neural networks, it would have also been better to present results from neural network with different layer widths.

**Questions:**

- Are there other tasks that the method can play an important part? For example, change/defect detection within point cloud data may be one direction of possibilities, but as the method requires taking the mean of all the elements, I am presuming it is rather difficult.

- What are the relationship between the proposed kernel and the network layers? Does the metric become more accurate as the width and number of layers increase? The authors do conduct analysis by changing the layers and comparing the final output, but I am curious about how relationship among shapes change according to the network structure.

- As the method states, the NTK would be identical if the network tends to infinity, but as this is unrealistic, I believe there needs to be some practical solution to the network design. What justifies the selection of the current architecture?

**Details Of Ethics Concerns:**

I believe there is little concern in terms of ethics, therefore I believe no further ethics review is necessary.

---

> ### Author Response · Authors · 2023-11-20
> **Reply to Reviewer 6p7s's Weakness 1**
>
> ##  •  Despite the very interesting theoretical approach of using varifolds, its practicality remains questionable. The interesting results can be found in small-sample shape classification tasks, where the proposed method and representation outperformed other conventional methods. However in most other tasks, the method had been outperformed by conventional methods with presumably more expensive computation. Despite the authors’ claim that the representation is able to extract both global and local shape similarities, the experiments demonstrate otherwise. It would have been better if the authors proposed a specific network structure that is able to take better advantage of the NTK representation. In order to compensate for the practical disadvantages, the authors could have introduced theoretical advantages over the conventional approaches. Such seems to be missing in the paper.
>
> We appreciate the reviewer's remark on the manuscript. To clear some misunderstanding regarding the computational complexity and methods' performance (and avoid redundant response), we would also like to refer the reviewer to our reply to reviewer wzuY. Please see below for our further response.
>
> Only for shape classification with all data, the performance of the neural varifold is not as comparable as the existing baselines like PointNet, DGCNN or more advanced methods, indicating the potential current limitation of the proposed methods in this specific case.  We would like to highlight that the performances of the proposed approaches on shape classification with  limited data, shape reconstruction and shape matching are either  better in many cases or at least comparable with other methods.
>
> In the limited data scenario, the proposed methods are actually not computationally heavier than neural networks with respect to the training time. For example, the computational time for 5-sample training on ModelNet10 is respectively 47 and 18 seconds for PointNet-NTK1 and PointNet-NTK2, while it takes 254 and 502 seconds for training PointNet and DGCNN with a single NVIDIA 3090 GPU, respectively. Although the inference time for the standard neural network could be cheaper after training completed, considering the performance gap and computational cost for our proposed methods, it is still within acceptable range and is worth improving further.
>
> In the case of shape reconstruction, Table 3 in the manuscript shows that none of the baseline methods are dominating the task and that the proposed method is competitive to the baseline methods. Note that we have also included a SOTA method -- neural kernel surface reconstruction (NKSR) -- into baseline in Table 3 in our revised manuscript. In addition, we have added more visualisation of the shape reconstruction results from all baseline methods as well as our proposed method. In the case of shape matching, we will explain in more detail in our response to your Weakness 2 below.

---

> ### Author Response · Authors · 2023-11-20
> **Reply to Reviewer 6p7s's Weakness 2**
>
> ## • The non-rigid registration results are very interesting, however, only two samples were provided for the experiments. It would be more convincing to apply the method on various shapes to analyze the tendencies of the proposal. Therefore, I believe the experiments are incomplete.
>
> In our revised manuscript (which will be uploaded to OpenReview before the deadline), we have incorporated three additional shape matching results (i.e., 1. two hippocampi, 2. sphere to Stanford Bunny, and 3. two airplanes) into the Appendix, outlined in the table below  as a showcase firstly.
>
> |             | Metric  | Chamfer | EMD     | CT                | NTK1             | NTK2             |
> |-------------|---------|---------|---------|-------------------|------------------|------------------|
> | Hippocampus | Chamfer | 3.49E-1 | 3.2E-1  | **2.43E-1**  | 2.67E-1          | 2.65-1           |
> |             | EMD     | 2.80E5  | 2.10E5  | 2.25E5            | 2.09E5           | **1.96E5**  |
> |             | CT      | 2.27E3  | 2.92E5  | 2.32E3            | 2.19E3           | **2.15E3**  |
> |             | NTK1    | 1.84E5  | 1.01E9  | 59.7E5            | **4.93E3**  | 9.98E3           |
> |             | NTK2    | 6.37E4  | 3.09E9  | 1.56E6            | **1.54E3**  | **1.54E3**  |
> | Bunny       | Chamfer | 9.32E-3 | 5.12E-3 | **3.60E-3**  | 4.40E-3          | 4.32E-3          |
> |             | EMD     | 2.31E4  | 4.74E3  | 3.72E3            | **3.13E3**  | 3.52E3           |
> |             | CT      | 2.40E-1 | 1.25E0  | **7.51E-2**  | 1.28E-1          | 1.23E-1          |
> |             | NTK1    | 2.57E-2 | 1.32E-2 | 1.83E-3           | **2.22E-4** | 2.94E-4          |
> |             | NTK2    | 3.85E-2 | 2.68E-2 | 3.33E-3           | 8.85E-4          | **6.43E-4** |
> | Airplane    | Chamfer | 1.36E-3 | 4.07E-4 | **3.72 E-3** | 3.81E-3          | 5.90E-3          |
> |             | EMD     | 1.16E4  | 4.12E2  | **3.38E2**   | 3.43E2           | 7.50E2           |
> |             | CT      | 8.71E-2 | 3.62E0  | **-3.58E-4** | 1.67E-3          | 3.68E-3          |
> |             | NTK1    | 2.27E-3 | 1.80E-1 | 0.41E-6           | **0.31E-6** | 0.72E-6          |
> |             | NTK2    | 6.14E-2 | 5.13E0  | 8.69E-6           | 3.17E-6          | **2.42E-6** |
>
> In all three instances, both NTK1 and NTK2 exhibit excellent results. In the case of shape matching between two hippocampi, NTK2 demonstrates superior results in terms of EMD, CT, and NTK2 metrics; for shape matching with NTK1, the best results are observed with respect to NTK1 and NTK2 metrics. For the Stanford Bunny, NTK1 yields the best outcomes in terms of EMD and NTK1 metrics; and the shape matching with NTK2 achieves optimal results with respect to the NTK2 metric. For shape matching between two airplanes, CT achieves the best results in terms of Chamfer, EMD and CT metrics, while both NTK1 and NTK2  achieve the best outcomes with respect to themselves. In particular, the visualisation of the airplane matching results shows that none of the methods match the target shape perfectly; and qualitatively, the shape matching results between airplanes are in favour of our methods NTK1 and NTK2 with respect to wings and fuselage. The detailed  visualisation of the additional experimental results has been added into the Appendix in the revised manuscript, which will be uploaded to OpenReview before the authors-reviewers discussion deadline.

---

> ### Author Response · Authors · 2023-11-20
> **Reply to Reviewer 6p7s's Weakness 3**
>
> ## •   Some ablation study had been conducted in the supplementary material by changing the number of layers of the target network. However, as the neural tangent kernel is derived from the study that links kernels to over-parameterized neural networks, it would have also been better to present results from neural network with different layer widths.
>
>
> We appreciate the reviewer's suggestion. In response to this suggestion, we have conducted experiments on the 9-layer MLP-based PointNet-NTK2, varying the width settings to include 128, 512, 1024, 2048, and infinite-width configurations. The new experiments will be added to the Appendix of the revised manuscript. We trained the model on 5 randomly sampled point clouds per class from the ModelNet10 training set. The evaluation was carried out on the ModelNet10 validation sets. This process was repeated five times with different random seeds, and the average shape classification accuracy was computed. Notably, PointNet-NTK1 was excluded from this experiment due to the absence of a finite-width neural network layer corresponding to the elementwise product between two neural tangent kernels of infinite-width neural networks. The results presented in the table below demonstrate that the analytical NTK (infinite-width NTK) outperforms the empirical NTK computed from the corresponding finite-width neural network with fixed layer width sizes.
>
> | Number of width for each layer | PointNet-NTK2 (5-sample) |
> |--------------------------------|--------------------------|
> | 128-width                      | 78.74 $\pm$ 3.30         |
> | 512-width                      | 80.08 $\pm$ 3.02         |
> | 1024-width                     | 79.97 $\pm$ 3.24         |
> | 2048-width                     | 80.46 $\pm$ 3.13         |
> | infinite-width                  | 81.74 $\pm$ 3.16  |
>
>
> Furthermore, computing empirical neural tangent kernels with respect to different length of width is known to be expensive since the empirical NTK is expressed as the outer product of the Jacobians of the output of the neural network with respect to its parameters. The details of the computational complexity and potential acceleration have been studied in [1]. However, if the finite-width neural networks are trained with the standard way instead of using empirical NTKs on the CIFAR10 dataset (containing 60,000 images), then finite-width neural networks outperform the neural tangent regime with performance significant margins [2,3]. In other words, there is still a large gap understanding about training dynamics of the finite-width nueral networks and their empirical neural kernel representations.

---

> ### Author Response · Authors · 2023-11-20
> **Reply to Reviewer 6p7s's Question 1**
>
> ## • Are there other tasks that the method can play an important part? For example, change/defect detection within point cloud data may be one direction of possibilities, but as the method requires taking the mean of all the elements, I am presuming it is rather difficult.
>
> Let us firstly clear a misunderstanding. Taking the mean of all elements in varifold kernels between two shapes is necessary for some applications (e.g. shape classification and shape matching), but this is not always the case (e.g. shape reconstruction). In the case of shape reconstruction, we generate off surface points by adding or deducting a constant $\delta$, and then assign labels per point. If points are on the surface, we then assign the label zero. If points are outside of the surface, we then assign either $\delta$ or $-\delta$ depending on the direction of the off surface points. Afterwards, we do the standard implicit surface reconstruction using the kernel regression. The changes of the shape/defect detection in theory is possible. However, it might not be scalable as it requires a prohibitively large kernel when the number of shapes increases.
>
> The varifold framework is flexible enough to integrate additional information on the surface, in addition to positions and normals. Thanks to the measure-theoretic structure, this can be done very easily using tensor product of measures. For example, Equation (32) in [5] defines functional varifolds that can take into account texture information as a function. In our context, this opens the way to the use of the varifold structure for handling shapes with texture information for various tasks, e.g., texture analysis, shape registration, shape reconstruction with texture, etc.

---

> ### Author Response · Authors · 2023-11-20
> **Reply to Reviewer 6p7s's Question 2**
>
> ## • What are the relationship between the proposed kernel and the network layers? Does the metric become more accurate as the width and number of layers increase? The authors do conduct analysis by changing the layers and comparing the final output, but I am curious about how relationship among shapes change according to the network structure.
>
>  As the reviewer requested, the behavior of the NTK pseudo-metrics with respect to the number of layers has been evaluated and will be added to the Appendix of the revised manuscript. Note that the width is not considered as all pseudo-metrics are computed analytically (i.e., infinite-width) here; please refer to our response to your previous question regarding the property of the proposed method with different layer width. In this study, simple shape matching networks were trained solely by NTK psuedo-metrics with different number of layers.
>
> The table  below showcases that the shape matching network trained with 5-layer NTK1 metric achieved the best score with respect to Chamfer, EMD, CT and NTK2 metrics, while the one with 9-layer NTK1 metric achieved the best score with respect to CT and NTK1 metrics. This is in accordance with the ablation analysis for shape classification, where 5-layer NTK1 achieved the best classification accuracy in the ModelNet10 dataset.  NTK2, on the other hand, shows a mixed signal. The shape matching network trained with the 1-layer NTK2 metric achieved the best outcome with respect to Chamfer, CT and NTK2 metrics, while the one trained with 9-layer NTK2 achieved the best results with respect to EMD and CT metrics. The network trained with 5-layer NTK2 only showed the best result with respect to NTK1 metric. This is not exactly in accordance with respect to shape classification with NTK2 metric, where the shape classification accuracy improves as the number of layers increases.  However, training a neural network always involves some non-deterministic nature; therefore, it is yet difficult to conclude whether the number of neural network layers is important for improving the shape quality or not.
>
> | Metric  | NTK1 (1)         | NTK1(5)          | NTK1(9)         |
> |---------|------------------|------------------|-----------------|
> | Chamfer | 2.82E-1          |  **2.67E-1** | 2.99E-1         |
> | EMD     | 2.43E5           | **2.09E5**  | 2.46E5          |
> | CT      | 2.19E3           | **2.17E3**  | **2.17E3** |
> | NTK1    | 7.74E3           | 4.93E3           | **4.90E3** |
> | NTK2    | 2.56E3           | **1.54E3**  | 1.92E3          |
> | **Metric**  | **NTK2 (1)**         | **NTK2 (5)**         | **NTK2 (9)**        |
> | Chamfer | **2.59E-1** | 2.61E-1          | 2.64E-1         |
> | EMD     | 2.14E5           | 2.32E5           | **1.93E5** |
> | CT      | **2.15E3**  | 2.17E3           | **2.15E3** |
> | NTK1    | 9.57E3           | **8.70E3**  | 9.98E3          |
> | NTK2    | **1.28E3**  | 1.41E3           | 1.53E3          |

---

> ### Author Response · Authors · 2023-11-20
> **Reply to Reviewer 6p7s's Question 3**
>
> ## • As the method states, the NTK would be identical if the network tends to infinity, but as this is unrealistic, I believe there needs to be some practical solution to the network design. What justifies the selection of the current architecture?
>
> The current architecture is based on one of the most popular neural network PointNet for 3D point clouds. The main idea is that the given network needs to be permutation invariant (e.g., fully connected layer, 1D convolution with 1-width convolution window, and average pooling). Even though our proposed methods have achieved excellent results in several important tasks for 3D point clouds, the proposed design shared the drawback from the adopted PointNet architecture and thus local variations cannot be able to be captured very efficiently as local neighbourhood is not considered in the architecture. The extension of the current work on graph neural tangent kernel is promising to address this case. We will try to see if we could add experiments in this regard within the limited authors-reviewers discussion period (note that we have added a significant number of new experiments in the revised manuscript); otherwise, we will leave it for future study since it is worth being investigated separately and deeply.

---

> ### Author Response · Authors · 2023-11-20
> **References**
>
> [1] Novak, Roman, Jascha Sohl-Dickstein, and Samuel S. Schoenholz. "Fast finite width neural tangent kernel." International Conference on Machine Learning. PMLR, 2022.
>
> [2] Arora, Sanjeev, et al. "On exact computation with an infinitely wide neural net." Advances in neural information processing systems 32 (2019).
>
> [3] Lee, Jaehoon, et al. "Finite versus infinite neural networks: an empirical study." Advances in Neural Information Processing Systems 33 (2020): 15156-15172.
>
> [4] Du, Simon S., et al. "Graph neural tangent kernel: Fusing graph neural networks with graph kernels." Advances in neural information processing systems 32 (2019).
>
> [5] Charlier, Benjamin, Nicolas Charon, and Alain Trouvé. "The fshape framework for the variability analysis of functional shapes." Foundations of Computational Mathematics 17 (2017): 287-357.

---

> ### Comment · Reviewer_6p7s · 2023-11-23
> **Thank you**
>
> Thank you for thoroughly addressing the questions and concerns.
> As shape classification may not be the strength of this proposal, (as this led to every reviewer in believing that the overall performance was rather subpar), I think the paper could be reformatted by allocating more space for the experiments added during rebuttal on the main manuscript.

---

> > ### Author Response · Authors · 2023-11-23
> > **Reply to Reviewer 6p7s -- reformatting some of the shape classification results, etc.**
> >
> > We greatly appreciate again your meticulous review of our work and your invaluable suggestion. Regarding your suggestion on reformatting the experiments to emphasise the great performance of our methods, we quite agree and have been working continuously on it following your suggestion. For example, part of the shape classification results will be moved to Appendix so that more space will be available to add some new results about shape matching (which will help to clear concerns regarding visualisation as well) in the main content of the manuscript.
> >
> > We will upload our revised manuscript soon before the authors-reviewers discussion deadline in a few hours.

---

> > ### Author Response · Authors · 2023-11-23
> > **Reply to Reviewer 6p7s -- reformatting some of the shape classification results, etc.**
> >
> > We greatly appreciate again your meticulous review of our work and your invaluable suggestion. Regarding your suggestion on reformatting the experiments to emphasise the great performance of our methods, we quite agree and have been working continuously on it following your suggestion. For example, part of the shape classification results will be moved to Appendix so that more space will be available to add some new results about shape matching (which will help to clear concerns regarding visualisation as well) in the main content of the manuscript.
> >
> > We will upload our revised manuscript soon before the authors-reviewers discussion deadline in a few hours.

---

### Official Review · Reviewer_wzuY · 2023-11-01

**Soundness:** 3 good
**Presentation:** 3 good
**Contribution:** 2 fair
**Rating:** 5
**Confidence:** 3

**Summary:**

The paper combines the concepts of varifolds from geometric measure theory defined for point clouds with neural tangent kernels (NTK) for infinite-width MLPs. Since the initial point cloud neural network architecture PointNet is an MLP, the main results of NTK directly translate to it. The authors propose to consider two variants of neural varifolds for point clouds constructed with: (1) product of separate NTKs for point coordinates and surface normals; (2) a single NTK for extended R^6 space of point and normal coordinates.

Using the resulting kernel methods, the authors specify how to use kernel ridge regression to solve shape classification and shape reconstruction, and also perform shape matching between point cloud pairs by minimization of a pseudo-metric for varifolds.

In the experiments, the authors evaluate the proposed methods and show that: 1) classification quality can not reach the performance of a standard PointNet network, although it can be competitive in a few-shot training setup; 2) shape reconstruction is competitive to some baselines; 3) shape matching experiment through varifold metric minimization shows promising results.

**Strengths:**

I am not a specialist on this topic, but to my knowledge, it is the first work to view PointNet-like architectures for point clouds as NTK approach. Additional connection to varifolds looks novel and promising as it is possible that some further results from the geometric measure theory could be explored in the context of 3D shape analysis.

**Weaknesses:**

1. Given that the complexity of the approach is higher compared to regular trainable networks and that the performance in the applications is lacking it is hard to find arguments in favour of using the proposed approach in practice.

2. Few-shot 3D point cloud classification is an established task and it would be much more relevant to compare to approaches designed to few-shot setup instead of regular classification methods: A Closer Look at Few-Shot 3D Point Cloud Classification.

3. Comparisons and proper positioning with respect to at least a couple of more recent baselines are missing: Neural Fields as Learnable Kernels for 3D Reconstruction, Neural Kernel Surface Reconstruction.

4. Additional visualizations of qualitative comparisons would be much appreciated.

**Questions:**

I am willing to improve my rating, but to be convinced I’d like to at least hear a couple of ideas of how the proposed method can be improved to close at least the performance gap or complexity gap (both, if possible).

---
Post rebuttal:

First of all, I would like to thank the authors for all the responses and clarifications. I appreciate the additional results added and all the discussions about possible improvements to the proposed method. Unfortunately, I still find it hard to improve my recommendation, since I believe this work still lacks either a (1) deeper connection to the concept of varifolds or (2) more convincing experimental results.

Regarding (1), the authors provide a link to this concept and never expand beyond using definitions to assign additional meaning to the model components that they use. It would have been more convincing if they had provided some additional implications allowed by the varifold framework.

Regarding (2), I still think that the results of the small dataset classification experiment are not convincing. After clarification from the authors, I agree that their setup is not the same as a standard few-shot classification setup. However, there are two aspects I'd like to mention. First of all, I think this small dataset setup is artificial and not practical. ShapeNet and even much larger Objaverse datasets are publicly available and we do not need to limit ourselves to its small subsets in practice. Maybe it makes sense to use another dataset (that is small without cutting its parts) to conduct this experiment. Secondly, even if we allow for this small dataset setting, the baselines for comparisons should be different. Instead of training the networks on the same amount of data, the authors can compare their models to pre-trained models in the zero-shot setting. Recent zero-shot baselines (Uni3D: Exploring Unified 3D Representation at Scale) show 88.2% zero-shot classification accuracy on ModelNet40 while the proposed approaches never reach such high accuracies even for 50 per-class training samples. These two aspects make it hard to justify the approach from a practical point of view at least in the classification setting.

Nevertheless, I appreciate the insightful ideas in this work and encourage the authors to continue to improve this submission before the next submission.

---

> ### Author Response · Authors · 2023-11-20
> **Reply to Reviewer wzuY's Weakness 1 & Question (Part1)**
>
> ## • Given that the complexity of the approach is higher compared to regular trainable networks and that theperformance in the applications is lacking it is hard to find arguments in favour of using the proposed approach in practice.
>
> ## • a couple of ideas of how the proposed method can be improved to close at least the performance gap or complexity gap (both, if possible)
>
> The computational complexity of the neural kernels (e.g., the neural tangent kernel and neural Gaussian process) has been a well known problem. A number of studies have been investigated to accelerate the kernel computation speed in terms of parallelisation as well as kernel approximations. The framework used in this study ``neural tangents'' [1] provides a powerful GPU parallelism with batch processing. Similarly, KeOps [2] also provides similar features such that it can accelerate kernel computation with GPU parallelism. However, it is still not able to solve the fundamental problem about its quadratic complexity. Recent studies have more focused on reducing the computational complexity by approximating neural kernels. For example, the neural kernels with ReLU activation can be approximated by a variant of linear sketch algorithms [3]. Table 1 in [3] shows that the computation of the convolutional neural tangent kernel (CNTK) for 3-layer 2D convolutional network with the proposed method (5,160 seconds) is around 200 times faster than the exact computation of CNTK ($>$1,000,000 seconds) on the CIFAR10 dataset. Further generalisation of the neural kernel approximation with various nonlinear activation functions has been introduced in [4]. In [4], the proposed kernel computation method (1.4 GPU hours) achieved also 106 times faster than the exact computation (151 GPU hours) of CNTK for Myrtle-5 on the CIFAR dataset. Furthermore, the performance of the approximated neural kernel is indeed similar or even better than the performance of the exact neural kernel [3,4]. The proposed approaches [3] and [4] can be directly translated and modified to both PointNet-NTK1 and PointNet-NTK2 representations, which may accelerate the kernel computation speed without loss of performance. We will leave it for future study.
>
> In the empirical point of view, the computational complexity of the PointNet-NTK1 and -NTK2 computation is actually not a significant bottleneck for some applications. For example, the few-shot scenario shown in Table 2 in the manuscript, the computational cost for 5-sample training on ModelNet10 is respectively 47 and 18 seconds for PointNet-NTK1 and PointNet-NTK2, while it takes 254 and 502 for training PointNet and DGCNN epochs with a single 3090 GPU, respectively. The performance and time complexity  are comparable to the standard neural networks in the case of the limited data scenario. To clear any confusion regrading the computational complexity, we have added a revised paragraph (i.e., the last paragraph) in Section 4.1 in our revision.  In the case of shape reconstruction, most of the shape reconstruction can be completed within 5 seconds. Furthermore, it can be potentially further optimised by introducing KeOps framework like Neural Splines. In the case of shape matching, the computation of the neural varifold kernels is not a significant computational bottleneck in terms of shape evaluation purpose. However, if one needs to use the metric as a loss function to train the network, then it is slower than other metrics; therefore further optimisation is required for better scalability.

---

> ### Author Response · Authors · 2023-11-20
> **Reply to Reviewer wzuY's Weakness 1 & Question (Part2)**
>
> In order to miminise the performance gap, feature augmentation techniques can be explored. In the case of CIFAR10 classification, the work in [7] augmented the data by a factor of 20, and then used highly parallelised conjugate gradient method to solve the linear system and achieved 91.2\% accuracy, which is the current SOTA amongst kernel based methods. Although its performance is not as good as the SOTA neural networks, it can be further improved in the context of few-shot learning.
>
> Our architecture is based on a simple PointNet architecture. Therefore, it shares the drawback of the PointNet architecture that local variations cannot be able to be captured very efficiently as local neighbourhood is not considered in the architecture. It might be worth investigating graph neural tangent kernel and its variants [8] to take into account subtle variations within the local neighbourhood.  With the potential improvement of the computational complexity [3,4], it might be able to achieve better performance on few-shot learning, shape reconstruction and  shape matching. It is worth highlighting that the performance of the proposed methods PointNet-NTK1 and PointNet-NTK2 is actually quite promising (i.e., they perform either better or comparably with the SOTA methods) rather than lacking. Furthermore, various SOTA architectures like Point Transformer can be exploited for the improvement of the architecture side. We will leave it for future study.
>
> The general definition of a rectifiable varifold (see Definition 1) involves a multiplicity function $\theta$ that carries concentration/density information. In the current study, the multiplicity $\theta$ is defined as constant 1 for simplicity, but we consider future work investigating whether the multiplicity term can be learned by data-driven approaches like [6] to boost the performance. Furthermore, the varifold framework is flexible enough to integrate additional information on the surface, in addition to positions and normals. Thanks to the measure-theoretic structure, this can be done very easily using tensor product of measures. For example, Equation (32) in [9] defines functional varifolds that can take into account texture information as a function. In our context, this opens the way to the use of the varifold structure for handling shapes with texture information for various tasks (e.g. texture analysis, shape registration, shape reconstruction with texture, etc.).

---

> ### Author Response · Authors · 2023-11-20
> **Reply to Reviewer wzuY's Weakness 2**
>
> ## • Few-shot 3D point cloud classifi cation is an established task and it would be much more relevant to compare to approaches designed to few-shot setup instead of regular classification methods: A Closer Look at Few-Shot 3DPoint Cloud Classifi cation.
>
> We appreciate the reviewer's insightful remarks on few-shot learning. In the current experiment setting in the manuscript, it is difficult to directly apply the standard few-shot learning approaches. This is because the few-shot learning approaches in [5]  you suggest require pre-training on existing dataset. In the main experiment in [5], the backbone network was trained with 30 classes of the ModelNet40 dataset, and then few-shot algorithms were applied and evaluated on 10 classes of ModelNet40. Our current experiment, on the other hand, assumes that all 40 classes have limited samples (1, 5, 10 or 50 samples).  Since the code base of [5] is public, we have been trying hard to investigate whether more results with experimental setup in [5] (i.e., pretraining on 30 class and few-shot learning on 10 class) can be added within this authors-reviewers discussion time frame. However, it is an interesting direction for the expansion of the current work.

---

> ### Author Response · Authors · 2023-11-20
> **Reply to Reviewer wzuY's Weakness 3 & 4 (Part1)**
>
> ## • Comparisons and proper positioning with respect to at least a couple of more recent baselines are missing: NeuralFields as Learnable Kernels for 3D Reconstruction, Neural Kernel Surface Reconstruction.
> ## • Additional visualizations of qualitative comparisons would be much appreciated.
>
> | Metric       | Method         | Airplane       | Bench          | Cabinet        | Car            | Chair          | Display        | Lamp            | Speaker        | Rifle          | Sofa           | Table          | Phone          | Vessel         |
> |--------------|----------------|----------------|----------------|----------------|----------------|----------------|----------------|-----------------|----------------|----------------|----------------|----------------|----------------|----------------|
> | CD (mean)    | SIREN          | 1.501          | 1.624          | 2.430          | 2.725          | **1.556** | **2.193** | 1.392           | 7.906          | 1.212          | 1.734          | 1.856          | **1.478** | 2.557          |
> |              | Neural Splines | 4.145          | **1.304** | 1.969          | 2.131          | 1.828          | 4.577          | **1.062**  | **2.798** | **0.400** | **1.650** | **1.576** | 10.058         | 2.210          |
> |              | NKSR           | 1.141          | 2.000          | 2.423          | 2.198          | 2.520          | 17.720         | 5.477           | 3.622          | 0.414          | 1.848          | 2.493          | 1.547          | 1.093          |
> |              | NTK1           | **0.644** | 1.314          | **1.991** | **2.107** | 1.734          | 4.666          | 1.134           | 2.806          | 0.425          | 1.654          | 1.586          | 10.397         | **1.079** |
> | CD (Median)  | SIREN          | 0.733          | 1.384          | 2.153          | 2.134          | 1.230          | 1.469          | 0.661           | 3.304          | 0.581          | 1.706          | 1.670          | 1.424          | 1.112          |
> |              | Neural Splines | 0.947          | 1.289          | 1.799          | **1.640** | **1.160** | **1.413** | **0.479**  | **2.749** | 0.347          | 1.586          | 1.372          | 1.600          | **0.788** |
> |              | NKSR           | 1.205          | 1.426          |**1.797** | 1.830          | 1.236          | 1.565          | 1.579           | 2.945          | **0.326** | 1.638          | 1.637          | **1.305** | 0.894          |
> |              | NTK1           | **0.621** | **1.259** | 1.828          | 1.836          | 1.237          | 1.499          | 0.566           | 2.794          | 0.352          | **1.578** | **1.350** | 1.558          | 0.797          |
> | EMD (mean)   | SIREN          | **2.990** | 3.763          | 4.983          | 5.208          | **4.649** | **4.658** | 24.068          | 13.292         | 2.418          | **3.688** | 8.745          | **3.237** | 4.500          |
> |              | Neural Splines | 22.004         | **3.571** | **4.420** | **4.694** | 7.916          | 9.205          | **16.786** | **5.857** | **1.503** | 3.706          | **4.194** | 17.846         | 5.957          |
> |              | NKSR           | 7.153          | 8.456          | 8.018          | 8.190          | 16.824         | 31.182         | 21.182          | 9.984          | 2.329          | 5.871          | 13.658         | 4.152          | 4.581          |
> |              | NTK1           | 3.120          | 4.153          | **4.420** | 4.767          | 7.350          | 9.653          | 23.381          | 6.236          | 1.592          | 3.888          | 5.259          | 24.101         | **3.534** |
> | EMD (median) | SIREN          | **2.690** | 2.938          | 4.520          | **3.803** | **4.411** | **3.314** | **2.279**  | 6.240          | 1.605          | 3.653          | 3.782          | **3.060** | 2.576          |
> |              | Neural Splines | 6.873          | **3.068** | **4.154** | 3.999          | 4.740          | 4.053          | 3.802           | **5.123** | **1.216** | **3.543** | **3.695** | 3.838          | 2.210          |
> |              | NKSR           | 5.732          | 5.119          | 4.440          | 5.313          | 5.683          | 3.777          | 4.927           | 5.975          | 1.227          | 3.641          | 6.375          | 3.088          | 2.771          |
> |              | NTK1           | 2.864          | 3.319          | 4.284          | 3.947          | 5.293          | 3.875          | 3.288           | 5.795          | 1.271          | 3.738          | 3.980          | 3.380          | **2.074** |

---

> ### Author Response · Authors · 2023-11-20
> **Reply to Reviewer wzuY's Weakness 3 & 4 (Part2)**
>
> As the reviewer recommended, we added another baseline -- Neural Kernel Surface Reconstruction (NKSR) [6] -- for comparison. We used the NVIDIA's pre-trained model available on the official GitHub repository of NKSR. The results are updated accordingly on figures and tables (e.g. see the table above a showcase of the comparison to NKSR) in the revised manuscript, which will be uploaded to  OpenReview before the deadline. As shown in the table above, NKSR only outperforms other methods with median Chamfer distance of the classes: Cabinet, Rifle and Phone. In the case of Neural Fields as Learnable Kernel for 3D reconstruction (NFLK), there is no public code base that is currently available. Due to the limited time frame for authors-reviewers discussion, it is not easily included into the baseline for comparison. However, since NKSR is indeed an extension of NFLK as stated in the Neural Kernel Surface Reconstruction [6], we hope you also agree adding NKSR for comparison is already sufficient and convincing.
>
> Furthermore, we will update the appendix of our manuscript to include more visualisation regarding the shape reconstruction results.

---

> ### Author Response · Authors · 2023-11-20
> **References**
>
> [1] Novak, Roman, et al. "Neural tangents: Fast and easy infinite neural networks in python." arXiv preprint arXiv:1912.02803 (2019).
>
> [2] Charlier, Benjamin, et al. "Kernel operations on the gpu, with autodiff, without memory overflows." The Journal of Machine Learning Research 22.1 (2021): 3457-3462.
>
> [3] Zandieh, Amir, et al. "Scaling neural tangent kernels via sketching and random features." Advances in Neural Information Processing Systems 34 (2021): 1062-1073.
>
> [4] Han, Insu, et al. "Fast neural kernel embeddings for general activations." Advances in neural information processing systems 35 (2022): 35657-35671.
>
> [5] Ye, Chuangguan, et al. "A Closer Look at Few-Shot 3D Point Cloud Classification." International Journal of Computer Vision 131.3 (2023): 772-795.
>
> [6] Huang, Jiahui, et al. "Neural Kernel Surface Reconstruction." Proceedings of the IEEE/CVF Conference on Computer Vision and Pattern Recognition. 2023.
>
> [7] Adlam, Ben, et al. "Kernel Regression with Infinite-Width Neural Networks on Millions of Examples." arXiv preprint arXiv:2303.05420 (2023).
>
> [8] Du, Simon S., et al. "Graph neural tangent kernel: Fusing graph neural networks with graph kernels." Advances in neural information processing systems 32 (2019).
>
> [9] Charlier, Benjamin, Nicolas Charon, and Alain Trouvé. "The fshape framework for the variability analysis of functional shapes." Foundations of Computational Mathematics 17 (2017): 287-357.

---

### Author Response · Authors · 2023-11-15
**Response to all the reviewers**

Thanks for your careful reading and appreciation of this work! All your questions and suggestions are of great help to us in improving the content and the presentation of this work.

We have updated the manuscript based on the reviewer's suggestions and advice. All changes are highlighted in red.

---

### Meta-Review · Area_Chair_MfL9 · 2023-12-06

**Metareview:**

A surface geometry characterization or a neural varifold representation of point clouds is proposed in this paper. Experiments show the effectiveness of the proposed method.

The paper received three “marginally below the acceptance threshold” ratings. All the reviewers think the experiments are insufficient and more convincing experiments are needed. Two reviewers expect more visualizations.

Reviewer wzuY thinks it is hard to find arguments in favour of using the proposed approach in practice and it would be much more relevant to compare to approaches designed to few-shot setup instead of regular classification methods. Moreover, Reviewer wzuY also thinks comparisons and proper positioning with respect to at least a couple of more recent baselines are missing. After rebuttals, Reviewer wzuY still find it hard to improve the recommendation because this work still lacks either a (1) deeper connection to the concept of varifolds or (2) more convincing experimental results.
The authors think it is unacceptable that review wzuY uses one reference ``Uni3D: Exploring Unified 3D Representation at Scale" that is under peer review in ICLR 2024 the same as our manuscript to conclude that our method cannot achieve better classification results. But，Uni3D is also a public paper: https://arxiv.org/abs/2310.06773.

Reviewer 6p7s thinks the practicality remains questionable, the experiments are incomplete, and it would have also been better to present results from neural network with different layer widths. After rebuttals, Reviewer 6p7s doesn't think the argument supporting the use of varifolds is too convincing. The overall improvements claimed in the paper isn't significant enough. Reviewer 6p7s still maintains the original score.

Reviewer vDG7 thinks the performance of neural varifolds is not state-of-the-art on most tasks, the network design in the manuscript is relatively simple, and more visualizations are favorable to highlight the characteristics and advantages of the proposed approach.

By observing the experimental results in the paper and in the rebuttals, the performances indeed are not state-of-the-art or the consistent best one. The paper still needs further revisions.

Based on the above comments, the decision was to reject the paper.

**Justification For Why Not Higher Score:**

The paper received three “marginally below the acceptance threshold” ratings. All the reviewers think the experiments are insufficient and more convincing experiments are needed. Two reviewers expect more visualizations.

Reviewer wzuY thinks it is hard to find arguments in favour of using the proposed approach in practice and it would be much more relevant to compare to approaches designed to few-shot setup instead of regular classification methods. Moreover, Reviewer wzuY also thinks comparisons and proper positioning with respect to at least a couple of more recent baselines are missing. After rebuttals, Reviewer wzuY still find it hard to improve the recommendation because this work still lacks either a (1) deeper connection to the concept of varifolds or (2) more convincing experimental results.
The authors think it is unacceptable that review wzuY uses one reference ``Uni3D: Exploring Unified 3D Representation at Scale" that is under peer review in ICLR 2024 the same as our manuscript to conclude that our method cannot achieve better classification results. I find Uni3D is a public paper: https://arxiv.org/abs/2310.06773.

Reviewer 6p7s thinks the practicality remains questionable, the experiments are incomplete, and it would have also been better to present results from neural network with different layer widths. After rebuttals, Reviewer 6p7s doesn't think the argument supporting the use of varifolds is too convincing. The overall improvements claimed in the paper isn't significant enough. Reviewer 6p7s still maintains the original score.

Reviewer vDG7 thinks the performance of neural varifolds is not state-of-the-art on most tasks, the network design in the manuscript is relatively simple, and more visualizations are favorable to highlight the characteristics and advantages of the proposed approach.

By observing the experimental results in the paper and in the rebuttals, the performances indeed are not state-of-the-art or the consistent best one. The paper still needs further revisions.

**Justification For Why Not Lower Score:**

N/A

---

### Decision · Program_Chairs · 2024-01-16

Reject